# Fibroblast A20 governs fibrosis susceptibility and its repression by DREAM promotes fibrosis in multiple organs

In addition to autoimmune and inflammatory diseases, variants of the TNFAIP3 gene encoding the ubiquitin-editing enzyme A20 are also associated with fibrosis in systemic sclerosis (SSc). However, it remains unclear how genetic factors contribute to SSc pathogenesis, and which cell types drive the disease due to SSc-specific genetic alterations. We therefore characterize the expression, function, and role of A20, and its negative transcriptional regulator DREAM, in patients with SSc and disease models. Levels of A20 are significantly reduced in SSc skin and lungs, while DREAM is elevated. In isolated fibroblasts, A20 mitigates ex vivo profibrotic responses. Mice haploinsufficient for A20, or harboring fibroblasts-specific A20 deletion, recapitulate major pathological features of SSc, whereas DREAM-null mice with elevated A20 expression are protected. In DREAM-null fibroblasts, TGF-β induces the expression of A20, compared to wild-type fibroblasts. An anti-fibrotic small molecule targeting cellular adiponectin receptors stimulates A20 expression in vitro in wild-type but not A20-deficient fibroblasts and in bleomycin-treated mice. Thus, A20 has a novel cell-intrinsic function in restraining fibroblast activation, and together with DREAM, constitutes a critical regulatory network governing the fibrotic process in SSc. A20 and DREAM represent novel druggable targets for fibrosis therapy.

Systemic sclerosis (SSc) is a complex autoimmune disease with heterogeneous clinical manifestations and varying clinical trajectories frequently culminating in irreversible organ failure[1,2]. Although the pathogenesis of SSc remains incompletely understood, considerable evidence implicates both genetic risk factors and environmental exposures[1]. While activated T and B cells, monocytes, macrophages, and dendritic cells are prominent in the early inflammatory stage of SSc, later stages of the disease are dominated by fibrosis synchronously affecting multiple organs[3]. The pathogenic networks of immune, vascular, and fibrotic processes fueling non-resolving organ fibrosis in SSc remain only partially characterized. Current views support critical roles for both cytokine-producing conventional immune cells and matrix-producing non-hematopoietic stromal cells[4].

Genome-wide association studies in SSc have uncovered multiple single-nucleotide polymorphisms (SNP) at the TNFAIP3 locus, encoding the ubiquitin-editing enzyme A20, that are strongly linked with disease susceptibility and fibrotic manifestations across multiple ethnic cohorts[5,6]. How these genetic variants are associated with altered A20 expression or function, potentially contribute to the disease process in SSc and in which cell types (immune versus non-immune), represent major unanswered questions. TNFAIP3 encodes the pleiotropic ubiquitin-editing enzyme A20, which has been linked to a diverse array of chronic autoimmune and inflammatory conditions[7,8]. A pivotal role of A20 is the suppression of NF-κB signaling to limit the intensity and duration of inflammatory responses in immune cells[7]. The absence of A20 in mice phenocopies multiple autoimmune and inflammatory conditions, while A20 loss-of-function mutations are

✉e-mail: bhattasw@med.umich.edu; vargaj@med.umich.edu

associated with severe immune and inflammatory diseases in humans[7,9].

While the expression, regulation, and mechanism of action of A20 have been extensively characterized in the context of inflammatory and autoimmune diseases linked to TNFAIP3 variants, little is currently known regarding A20 in the context of SSc, including its regulation and function in tissue-resident stromal cells and its potential pathogenic role in organ fibrosis. In view of the robust and consistent genetic linkage of SSc with TNFAIP3, here we sought to elucidate the potential contribution of A20 and its transcriptional regulator downstream regulatory element antagonist modulator (DREAM) to SSc pathogenesis. Using SSc patient-derived skin and lung tissues and isolated fibroblasts combined with experiments in engineered mice harboring germline or fibroblast-specific deletion of A20 or DREAM, we now demonstrate that A20 expression was reduced in SSc, while expression of DREAM, the negative regulator of A20, was significantly elevated and anti-correlated with A20. Mice with global loss-of-function of A20 in all tissue, or restricted to fibroblasts, demonstrated exaggerated inflammatory and fibrotic responses in skin and lung, while DREAM-null mice showed protection from fibrosis. Moreover, fibroblasts lacking DREAM showed enhanced A20 induction associated with near-complete attenuation of inducible fibrotic responses. An anti-fibrotic small molecule agonist of the adiponectin receptor elicited a sustained increase in A20 accumulation in isolated fibroblasts and in bleomycin-treated mice. Thus, we uncover a novel cell-intrinsic role for A20 in the negative regulation of profibrotic fibroblast responses. Impaired A20 expression or function in SSc patients due either to TNFAIP3 genetic variants and/or environmental regulatory influences resulting in altered DREAM expression might contribute directly to the development or progression of fibrosis in multiple organs and represents a novel target for anti-fibrotic therapy.

## Results

### Expression of A20 is downregulated in SSc

A schematic overview of the experimental pipeline is shown in Supplementary Fig. 1. Genome-wide association studies have uncovered significant and reproducible associations of multiple TNFAIP3 variants, encoding the ubiquitin-editing enzyme A20, with fibrotic SSc phenotypes[5]. To evaluate A20 expression in SSc, we initially queried a dataset comprising three distinct skin biopsy transcriptomes (GSE59785, GSE45485, and GSE32413) from a total of 76 SSc patients and 39 healthy controls[10]. SSc skin biopsies showed significantly reduced A20 transcript levels compared to healthy controls ($p = 0.016$) (Fig. 1A). No associations were detected between A20 expression levels in the SSc biopsies and disease duration, baseline modified Rodnan skin score (MRSS), or change in MRSS from baseline to 6 months. In healthy donor skin biopsies ($n = 6$) A20 was detectable primarily in the papillary dermis, while SSc skin biopsies ($n = 8$; Cohort 1, Table 1) showed significantly reduced numbers of A20+ fibroblasts ($p = 0.0007$), most notable in the papillary dermis (Fig. 1B). As expected, levels of procollagen I, a marker for activated fibroblasts, were elevated in these biopsies. Data from an independent second cohort (Cohort 2, $n = 6$–10) confirmed reduced A20 expression in SSc biopsies ($p = 0.0095$) (Supplementary Fig. 2 and Table 1). Moreover, levels of A20 were significantly lower in explanted SSc fibroblasts compared to healthy control fibroblasts ($n = 6$; $p = 0.0022$), accompanied by elevated levels of fibronectin-EDA ($n = 3$) and procollagen I ($n = 3$) (Fig. 1C). Levels of A20 mRNA were found to be reduced in SSc skin-derived fibroblasts ($n = 6$–10) in an independent transcriptome dataset[11], while expression of tenascin-C, Col1a1, Col1a2, Col15a1, Col15a2 and Col4a1, and other profibrotic genes were elevated (Table 2).

To evaluate alterations in A20 expression in the lung in SSc, we determined A20 mRNA levels in patients with SSc-ILD ($n = 16$) and non-SSc controls ($n = 5$)[12]. Levels of A20 mRNA were significantly reduced in

SSc-ILD lungs ($p = 0.0008$) (Fig. 1D). Immunolabeling of lung biopsies showed that A20 expression was primarily restricted to alveolar epithelial lining cells in non-SSc lungs ($n = 4$) and was significantly reduced in each SSc-ILD lung ($n = 7$) ($p = 0.0061$) (Fig. 1E). Analysis of transcriptome data of lung fibroblasts isolated from SSc-ILD biopsies ($n = 8$) showed significantly decreased in vitro expression of A20 mRNA ($p < 0.00001$) compared to control lung fibroblasts ($n = 10$) (Fig. 1F)[13]. Together, these findings indicate that A20 expression is downregulated in skin and lung biopsies, as well as in isolated skin and lung fibroblasts from patients with SSc.

### Exaggerated organ fibrosis in A20[+/−] mice

In view of the association between reduced A20 expression and skin and lung fibrosis in SSc, we sought to test the hypothesis that loss of A20 might play a direct pathogenic role in fibrosis. For this purpose, we generated A20 hypomorphic mice by crossing EIIa-Cre mice that carry a cre transgene under the control of the adenovirus EIIa promoter, and A20[fl/fl] mice, yielding mice with partial (~50%) reduction in A20 levels (Supplementary Fig. 3A), comparable to individuals harboring TNFAIP3 risk alleles[14]. In striking contrast to homozygous A20[-/-] mice that die perinatally due to severe multiple organ inflammation[15], mice that are haploinsufficient for A20 were viable, fertile, and without spontaneous phenotype for up to 6 months of age. While 2-month-old A20[+/−] mice showed no clinical or histological signs of inflammation, they demonstrated spontaneous IgG autoantibody production, unlike A20[fl/fl] mice of the same age and sex (Fig. 2A, left panel and Supplementary Data 1). Low-dose bleomycin treatment induced limited skin fibrosis in A20[fl/fl] mice, while identically treated A20[+/−] mice showed markedly exacerbated fibrotic responses, with significantly greater increases in dermal thickness ($p = 0.0043$) and Col1a1 and Col1a2 mRNA expression (Fig. 2B). To investigate the mechanisms underlying exaggerated fibrosis induction in A20[+/−] mice, we examined bleomycin-induced changes in skin transcript levels. At day 22 following initiation of bleomycin injections, a broad pattern of gene expression changes was noted in bleomycin-treated A20[+/−] mice, with >2-fold increase (916 genes) or decrease (762 genes) in transcript levels compared with untreated A20[+/−] mice ($P < 0.01$; false-positive rate (FDR) 0.05) (Fig. 2C and Supplementary Data 2A, B) (GSE194380). Intriguingly, several genes involved in fibrotic (Col8a1, Ostp, Timp1, S100a8, Pai1, Ctgf) and inflammatory (Saa3, Trem1, Trem3, Acod1, and Il1b) processes were upregulated in A20[+/−] mice (Supplementary Fig. 3B). Importantly, bleomycin-induced accumulation of activated (procollagen+) fibroblasts ($p = 0.0003$) and αSMA+ myofibroblasts ($p = 0.0007$) within the lesional dermis was accentuated in A20[+/−] mice compared to bleomycin-treated control mice, whereas the increase in the numbers of F4/80+ macrophages and perilipin+ adipocytes ($p = 0.90$ and $p = 0.128$, respectively) was not significantly different (Fig. 2E).

In order to elucidate potential molecular parallels in skin fibrosis between bleomycin-treated A20 haploinsufficient mice and SSc patients with reduced A20 expression, we conducted cross-species transcriptome analyses[10,16]. The majority of the genes showing differential expression in bleomycin-treated mice compared to untreated A20[+/−] mice correlated with their human counterparts differentially expressed in SSc patients ($p = 1.22e^{-18}$, Pearson correlation $r = 0.34$) (Fig. 2D). To further probe parallels in the gene expression changes in bleomycin-treated A20[+/−] mice and SSc patients, we performed the same cross-species analysis using a set of genes regulated by bleomycin in wild-type (A20[fl/fl]) mice. The results showed a smaller number of differentially expressed genes as well as reduced Pearson's correlation compared to A20[+/−] mice ($r = 0.23$; $p = 0.0046$). Thus, A20[+/−] mice show heightened susceptibility to the development of SSc-like fibrosis accompanied by pathological and genomic changes paralleling those seen in patients with SSc.

To explore the potential pathogenic role of A20 in a non-inflammatory fibrosis model, we generated A20[+/]; Tsk[1/+] mice. Due to a

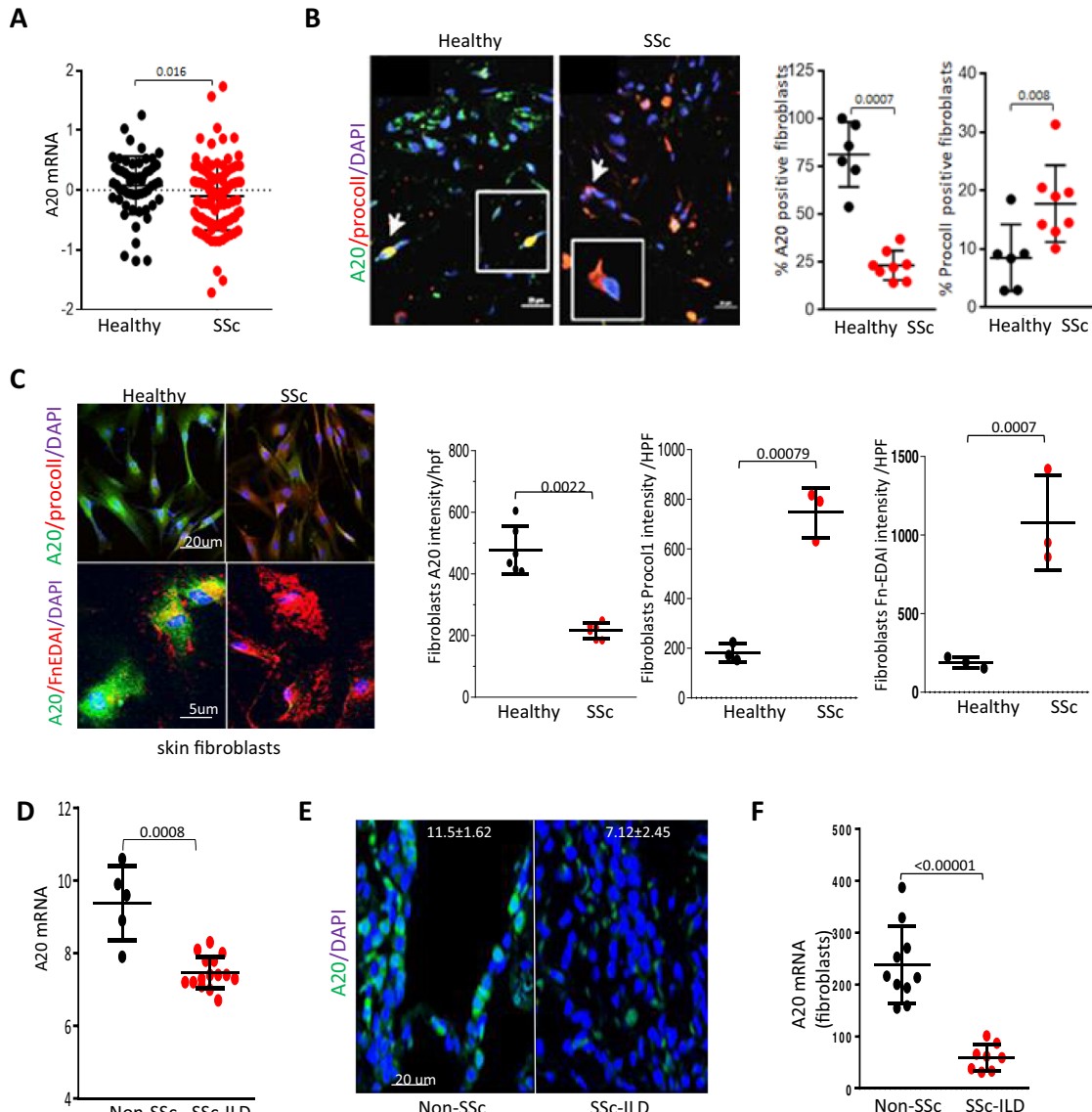

**Fig. 1 | Reduced A20 expression in SSc. A** A20 mRNA levels in SSc skin biopsy-derived transcriptome datasets (GSE59785, GSE45485 and GSE32413) (HC, $n = 39$, SSc, $n = 76$)[10]. Each dot represents a single subject. Two-tailed Mann−Whitney $U$ test. **B** SSc ($n = 8$) and healthy control ($n = 6$) skin biopsies were immunolabelled with antibodies to A20 or procollagen I (Procoll 1). Left panels: representative images. Inset: cells identified by arrows (higher magnification). Bar = 20 μm. Right panel: quantitation of immunofluorescence. The proportion of immunopositive cells in the dermis was determined in a blinded manner by scoring cells from four randomly selected hpf/slide. The numbers of immunopositive and -negative cells were used to calculate the percent of activated fibroblasts, A20+ cells, and A20+ activated fibroblasts. Two-tailed Mann−Whitney $U$ test. **C** Immunolabeling of confluent fibroblasts explanted from healthy ($n = 3$) or SSc ($n = 3$) skin biopsies. Left panel: representative images. Bars = 20 and 5 μm. Right panel: quantitation of immunofluorescence (each point represents mean intensity from four randomly selected hpf/slide). Two-tailed Mann−Whitney $U$ test. **D** A20 mRNA levels in the lungs from patients with SSc-ILD ($n = 16$) and non-SSc controls ($n = 5$)[12]. Two-tailed Mann−Whitney $U$ test. **E** SSc-ILD ($n = 7$) and non-SSc ($n = 4$) lungs were immunolabelled using antibodies to A20. Representative images; scale bar, 20 μm. $p = 0.0061$, Two-tailed Mann−Whitney $U$ test. **F** A20 mRNA levels from transcriptome data of lung fibroblasts isolated from SSc-ILD biopsies ($n = 8$) and healthy control ($n = 10$)[13]. Two-tailed Mann−Whitney $U$ test.

fibrillin-1 gene tandem duplication mutation, Tsk1/+mice spontaneously develop fibrosis of the skin in the absence of inflammation and vascular changes[17,18]. The characteristic increase in hypodermal thickness observed in 12-week-old Tsk1/+mice was significantly exacerbated in Tsk1/+mice lacking A20[+/−] ($p = 0.031$), while no inflammation in the skin was observed (Fig. 2F). Moreover, collagen accumulation on the skin was significantly augmented in A20[+/−]; Tsk[1/+] mice compared to Tsk[1/+] mice (Supplementary Fig. 3C).

To explore how A20 modulates the development of lung fibrosis, A20[+/−] mice and A20[fl/fl] control mice in parallel were treated with s.c. bleomycin for 2 weeks and sacrificed on day 22. Bleomycin-induced changes in the lungs, including the influx of inflammatory cells and

appearance of subpleural fibrotic foci, were markedly exaggerated in A20[+/−] mice (Fig. 3). In particular, pulmonary architectural distortion was exacerbated, with significantly greater fibrosis score ($p < 0.0001$) and interstitial collagen accumulation ($p = 0.038$). In addition, bleomycin-treated A20[+/−] mice showed a greater increase in *Col1a2* ($p = 0.023$) and *Il1b* ($p = 0.03$) mRNA levels, numbers of αSMA+ interstitial myofibroblasts, procollagen I+ activated fibroblasts and F4/80+ macrophages, and tissue deposition of the fibrosis marker tenascin-C (Fig. 3 and Supplementary Fig. 4).

A series of orthogonal experimental approaches were taken to characterize cellular pathways underlying exaggerated fibrosis in A20[+/−] mice. First, we examined differentially expressed genes in mice

**Table 1 | Clinical characteristics of SSc subjects**

| A20 detection by IF (dcSSc, early, *N* = 8) | | A20 detection by IHC (dcSSc, early, *N* = 6) | | DREAM measured by IF (dcSSc, early, *N* = 10) | |
|---|---|---|---|---|---|
| Age, mean ± SD | 55.38 ±11.66 | Age, mean ± SD | 54.33 ± 9.07 | Age, mean ± SD | 52.10 ±10.86 |
| Sex, *n* (% Female) | 75% | Sex, *n* (% Female) | 100% | Sex, *n* (% Female) | 60% |

Subjects providing skin biopsy samples for IF and IHC. Early, <3 years from the first non-Raynaud disease manifestation, late >3 years from the first non-Raynaud disease manifestation. Controls were healthy adults (75% female; median age: 50 years; range: 26–57 years).
*IF* immunofluorescence, *IHC* immunohistochemistry, *dcSSc* diffuse cutaneous systemic sclerosis systemic sclerosis.

**Table 2 | Gene expression in never-cultured skin fibroblasts[11] (Gardet et al., *Sci. Rep.* 2019, GSE TBD)**

| Gene symbol | Fold change (SSc vs HC) | FDR *p* value |
|---|---|---|
| TNFAIP3 | −2.87 | 0.001958 |
| TNC | 2.25 | 0.00821 |
| COL1A1 | 1.57 | 0.003018 |
| COL1A2 | 2.14 | 0.001764 |
| COL15A1 | 2.66 | 7.174E−06 |
| COL5A2 | 1.96 | 0.001364 |
| COL4A1 | 4.09 | 1.684E−06 |

Ten healthy subjects (HC) and six SSc patients with early diffuse cutaneous SSc (SSc) were studied.
To identify differentially expressed genes between groups of samples, authors applied the Linear Models for Microarray (LIMMA) data analysis. All *p* values for each probe were adjusted using false discovery rate (FDR <0.05) using the Benjamini and Hochberg method.

that were untreated or treated with bleomycin. KEGG Pathway Enrichment Analysis of differentially expressed genes in A20[+/−] mice revealed significant (*p* < 0.000001) enrichment of fibrotic pathways, including focal adhesion, Wnt, Hippo, and JAK-STAT signaling, and inflammatory pathways including NF-kB, TNF, and chemokine signaling as well as toll-like receptor (TLR)-mediated innate immune signaling in bleomycin-treated A20[+/−] mice (Supplementary Table 1).

In light of the potential role of Wnt signaling in SSc pathogenesis and our demonstration that mice with transgenic Wnt overexpression developed β-catenin-mediated skin fibrosis[19–25], we sought to compare β-catenin levels in A20[+/−] mice and A20[fl/fl] mice. Bleomycin-induced fibrosis was accompanied by significantly greater β-catenin accumulation in the skin (*p* = 0.028) and lung (*p* = 0.0045) in A20[+/−] mice compared to A20[fl/fl] mice, suggesting that canonical Wnt signaling activity was enhanced in the face of A20 deficiency (Supplementary Fig. 5).

Focal adhesion kinase (FAK) plays a fundamental role in mediating profibrotic cellular responses elicited by TGF-ß and other soluble and mechanical cues[26,27]. Significantly, FAK was shown to be constitutively activated in SSc fibroblasts, suggesting that it might be involved in pathogenesis[28,29]. To investigate if A20 modulates FAK activity, we studied mouse embryonic fibroblasts (MEFs) isolated from A20[-/-] and wild-type mice in parallel. Incubation of cultures with TGF-β elicited a significantly greater increase in FAK phosphorylation in A20[-/-] MEFs compared to wild-type MEFs (*p* = 0.027), identifying FAK as a molecular target of A20 in fibroblasts (Supplementary Fig. 6A).

Perhaps the most investigated A20 molecular target in the negative regulation of inflammatory responses is TRAF6, an adapter protein with ubiquitin ligase activity[30]. Upon TLR4 activation, TRAF6 undergoes K63-linked ubiquitination and recruits and activates TAK1 and IKK kinases. In contrast, A20, by removing ubiquitin chains from TRAF6, terminates NF-κB signaling and limits inflammation[31]. To examine if A20 modulates TRAF6 in the fibrotic context, we examined TRAF6 ubiquitination in A20[-/-] MEFs. Treatment with TGF-β elicited substantially greater TRAF6 ubiquitination and TAK1 activation in A20[-/-] MEFs compared to wild-type MEFs (*p* = 0.00036) (Supplementary Fig. 6B, C). Altogether, these results indicate that in fibroblasts, A20 negatively regulates fibrotic responses by targeting the activity of

distinct pathways (including TGF-ß, Wnt/ß-catenin, FAK, and TRAF6) that are implicated in the pathogenesis of SSc.

## Fibroblast A20 modulates fibrosis in the skin and lungs

In contrast to immune cells, very little is known regarding A20 expression, regulation, or mechanism of action in fibroblasts or its role in modulating mesenchymal cell responses. To determine if fibroblast A20 contributed to fibrosis, we crossed mice hemizygous for Cre (Col1a2-Cre[+/−]) with mice homozygous for loxP-A20 (A20[fl/fl]) to generate mice with fibroblast-specific A20 deletion (Supplementary Fig. 7, left panel). Fibroblast-selective A20 ablation upon tamoxifen injection in A20[fibcko] mice was confirmed in explanted skin and lung fibroblasts (Supplementary Fig. 7, right panel). A20[fibcko] mice showed no overt phenotype for up to 16 weeks of age and in contrast to A20[+/−] mice, had no evidence of increased humoral autoimmunity (Fig. 2A, right panel). Low-dose bleomycin treatment elicited a significantly greater increase in dermal thickness and skin collagen accumulation in A20[fibcko] mice compared to corn oil-treated A20[fl/fl] control mice (*p* = 0.027), accompanied by enhanced increase in *Col1a1, Col1a2, IL6*, and *Mcp1* mRNA expression (Fig. 4A, B) and in the numbers of αSMA+ myofibroblasts in the dermis (Fig. 4C).

Bleomycin injection in the skin elicited an influx of inflammatory cells in the lungs that was coupled with the emergence of fibrotic foci, architectural distortion, and increased collagen accumulation (Fig. 4D, E). Each of these bleomycin-induced pathological responses in the lung was notably exaggerated in A20[fibcko] mice, with significantly greater fibrosis score (*p* = 0.018) and increased interstitial collagen accumulation (*p* = 0.0065). Importantly, lung fibroblasts explanted from A20[fibcko] mice showed exaggerated ex vivo fibrotic responses when challenged with TGF-β (Supplementary Fig. 8).

## The downstream regulatory element antagonist modulator (DREAM) is upregulated in SSc patients and shows anti-correlation with A20

Originally identified as a neuronal Ca[2+]-sensing molecule implicated in pain modulation, DREAM, a 31-kDa cellular protein, was subsequently shown to also function as a transcriptional repressor with sequence-specific binding to the DRE downstream regulatory elements[32]. Significantly, we recently demonstrated that DREAM binds to the A20 promoter DRE to repress A20 expression in a variety of cell types[33]. We therefore surmised that aberrant A20 expression in SSc might result from its transcriptional repression mediated via DREAM. To evaluate this concept, we determined A20 and DREAM expression in the skin in biopsy-derived transcriptomes from multiple SSc datasets. We found that DREAM mRNA levels were significantly higher (*p* = 0.039) in SSc skin biopsies compared to healthy controls (Fig. 5A)[10]. Importantly, DREAM expression in these biopsies showed significant anti-correlation with A20 levels (*r* = −0.232; *p* = 0.011) (Fig. 5B). Elevated DREAM expression in the skin was confirmed by immunolabeling in SSc (*n* = 10) compared to healthy control (*n* = 10) biopsies (*p* = 0.0009) (Cohort 2, Table 2), with concomitant elevation in procollagen I expression in the dermis (Fig. 5C). Moreover, fibroblasts explanted from SSc skin biopsies (*n* = 6) showed both elevated DREAM expression (*p* = 0.0043), as well as increased nuclear localization (*p* = 0.011) in vitro (Fig. 5D). These

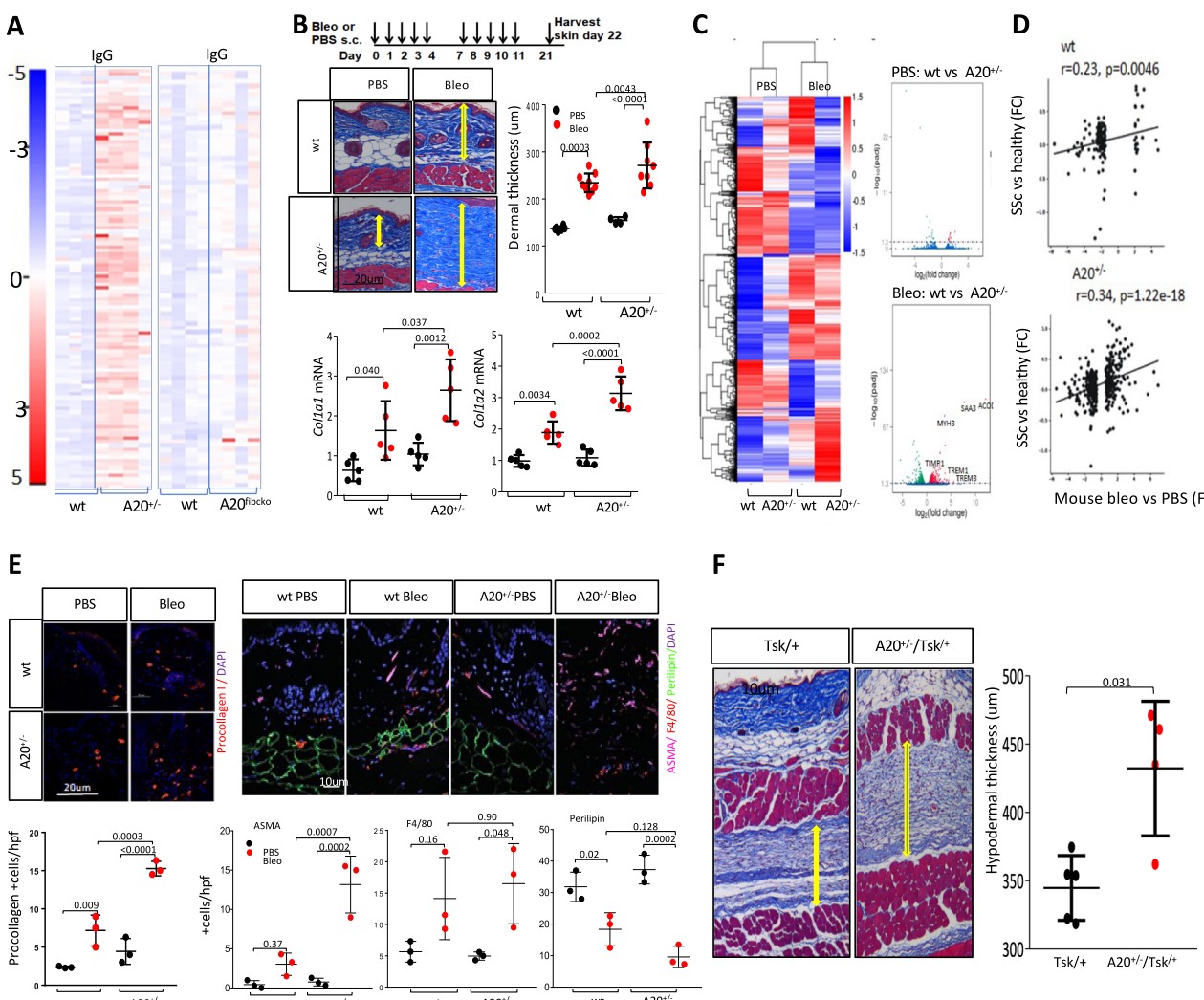

**Fig. 2 | Exaggerated skin fibrosis in A20⁺/⁻ mice.** A20^fl/fl mice and A20⁺/⁻ mice in parallel were treated with s.c. bleomycin (5 mg/kg) for 2 weeks and sacrificed on day 22. **A** Serum autoantibodies were examined using autoantigen arrays. Heatmap showing -fold change compared with the average in control samples. Heatmap of 112 IgG autoantibodies after filtered by SNR >3% across all samples. Red, upregulated; blue, downregulated. FDR = 0.05. **B** Top: schematic of the experiment. Left upper panels: lesional skin, Trichrome stain. Representative images. Bar = 20 μm. Arrows indicate dermis. Right upper panel: quantitation of dermis thickness (mean ± SD of eight determinations/hpf, A20^fl/fl PBS, $n = 6$, Bleo, $n = 9$; A20⁺/⁻, PBS, $n = 4$, bleo, $n = 8$). One-way analysis of variance followed by Sidak's multiple comparison test. Lower panels: qPCR of skin RNA. Results, normalized with GAPDH, are mean ± SD of triplicate determinations from three mice/group. One-way ANOVA followed by Sidak's multiple comparison test. **C** Skin transcriptome analysis. RNA isolated from the lesional skin was hybridized to Illumina human HT12 Microarray Chips. Left panel: heatmap showing -fold change in gene expression compared with the average in controls. Red, upregulated; blue, downregulated. FDR, 0.05, >1.5-fold change. Clusters indicate genes with expression significantly up- or

downregulated. Right panel: volcano plots of differentially up- or downregulated genes. **D** Two-sided Pearson's correlation of genome-wide RNA-seq-based cross-species comparison of gene expression between bleomycin vs PBS-treated A20⁺/⁻ mice and SSc patients vs. healthy subjects (y axis; $n = 72$ SSc, patients; $n = 36$ healthy subjects). **E** Skin biopsies were immunolabelled using antibodies to procollagen I (left upper panel, bar = 20 μm). ASMA, perilipin, and F4/80 (right upper panels, bar = 10 μm). Representative images. Lower panels: the proportion of immunopositive cells within the dermis determined at four randomly selected locations/hpf in each slide. ASMA+ cells with vessel morphology were excluded from analysis, and only ASMA+ interstitial cells with characteristic spindle-shaped morphology were counted. Results are mean ± SD from three mice/group. One-way ANOVA followed by Sidak's multiple comparison test. **F** Skin biopsies from 12-week-old male A20⁺/⁻; Tsk^1/+ and Tsk^1/+mice. Left panel: Trichrome stains. Arrows delineate the hypodermis. Bar, 10 μm. Representative images from 4 to 5 mice/group. Right panel: hypodermal thickness. Results are mean ± SD from 5 hpf/mouse. One-way ANOVA followed by Sidak's multiple comparison test.

results are the first to demonstrate altered DREAM expression in SSc and suggest that enhanced DREAM in SSc fibroblasts might be directly responsible for the reduced levels of A20 in a cell-autonomous fashion[34]. Importantly, knockdown of DREAM in SSc fibroblasts ($n = 5$) resulted in greater A20 expression that was accompanied by reduced Type I collagen levels and ASMA accumulation (Supplementary Fig. 9A). To investigate whether epigenetic mechanisms might potentially account for altered DREAM expression in SSc, we examined chromatin accessibility at the DREAM/KCNIP3 locus by extracting ChIP-seq data generated with human

dermal fibroblasts (ENCODE database). Significant enrichment with H3K27ac was noted at this locus (Supplementary Fig. 9B), suggesting that DREAM expression in skin fibroblasts might be under epigenetic regulation. To directly explore if histone acetylation might play a role in elevated DREAM expression in SSc, explanted skin fibroblasts ($n = 9$) were incubated with JQ1, a bromodomain and extra-terminal motif (BET) inhibitor that selectively binds to acetylated histones[35]. JQ1 treatment reduced the expression of DREAM (Supplementary Fig. 9C), supporting a potential role for histone acetylation in elevated DREAM expression in SSc fibroblasts.

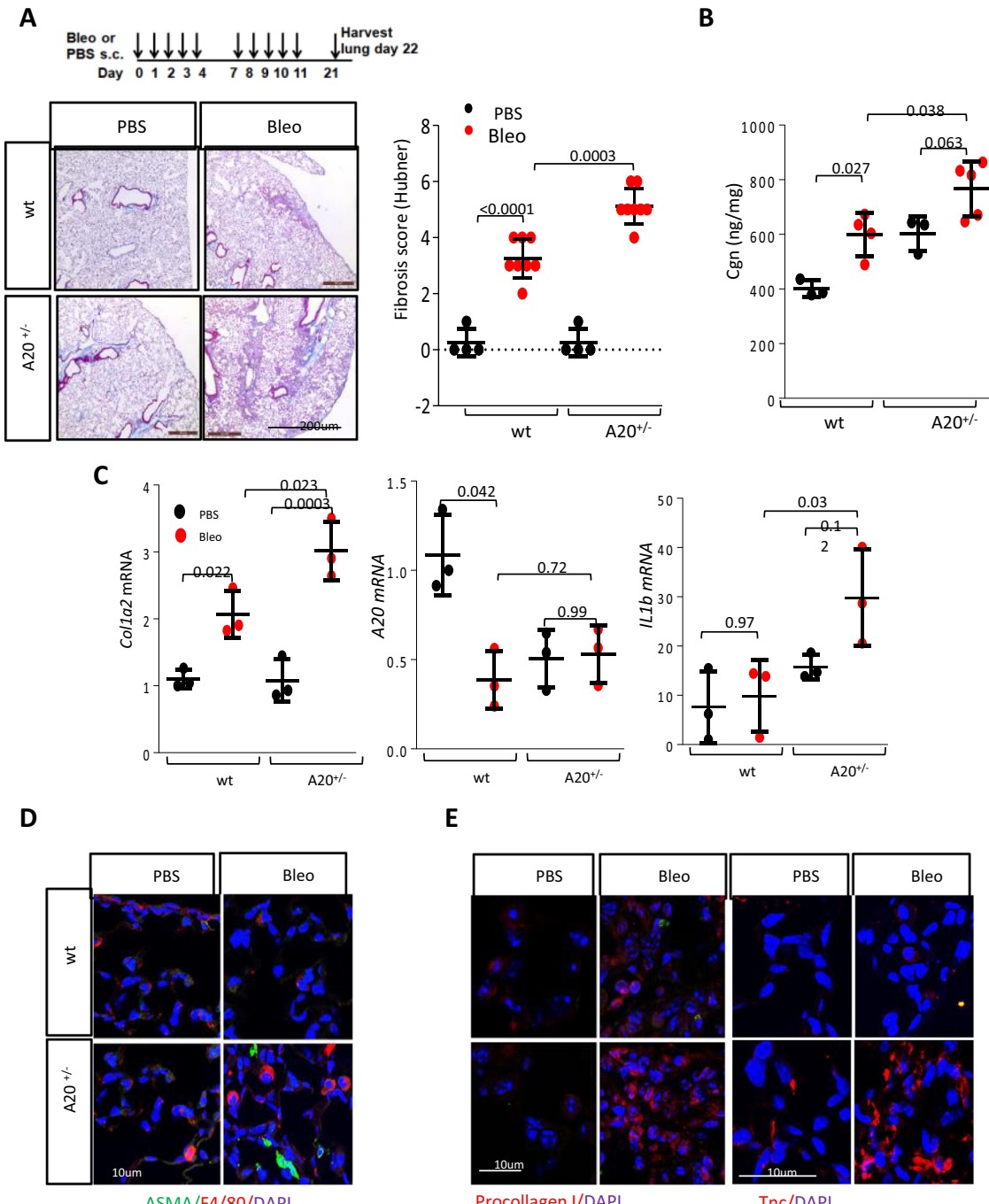

**Fig. 3 | A20⁺/⁻ mice showed exaggerated lung fibrosis.** Control A20^{fl/fl} (wt) mice and A20⁺/⁻ mice in parallel were treated with s.c. bleomycin (10 mg/kg) or PBS for 2 weeks, and lungs were harvested at day 22. **A** Top: schematic of the experiment. Left panel: Trichrome stains. Representative images; size bars = 200 μm. Right panel: lung fibrosis scores (Hubner); results are mean ± SD One-way ANOVA followed by Sidak's multiple comparison test (PBS-treated, n = 3; bleo-treated, wt, n = 4, KO, n = 5). **B** Lung collagen content: each dot represents the mean ± SD of duplicate determinations (PBS-treated, n = 4; bleo-treated, n = 7). One-way ANOVA followed by Sidak's multiple comparison test. **C** qPCR. Results, normalized with GAPDH, are mean ± SD of triplicate determinations from three mice/group. One-way ANOVA followed by Sidak's multiple comparison test. **D** Immunolabeling with antibodies to ASMA, F4/80 (scale bar = 10 μm). **E** Immunolabeling with antibodies to procollagen I and tenascin-C (scale bars = 10 μm). Representative images (n = 3).

## DREAM promotes fibrosis

Aside from its key roles in neuronal biology including regulation of pain[36,37], little is known about the potential extra-neuronal functions of DREAM. In view of the observed DREAM anti-correlation with A20 in SSc skin biopsies and DREAM's repressive effect on A20 expression in endothelial cells and SSc fibroblasts, we sought to test the hypothesis that DREAM might modulate profibrotic responses in the skin via repression of A20. For this purpose, we studied mice lacking DREAM. These mice were fertile, developed normally, and showed no overt behavioral changes or spontaneous phenotypes up to six months of age[32].

Bleomycin treatment was associated with a significantly attenuated increase in dermal thickness (p < 0.0001), collagen accumulation in the skin (p = 0.02), and *Col1a1*, *Col1a2*, and *ASMA* mRNA expression, in DREAM KO mice (Fig. 6A–C). Notably, dermal thickness showed significant anti-correlation with A20 expression in the lesional skin (r = −0.706, p = 0.013) (Fig. 6D) and was associated with elevated A20

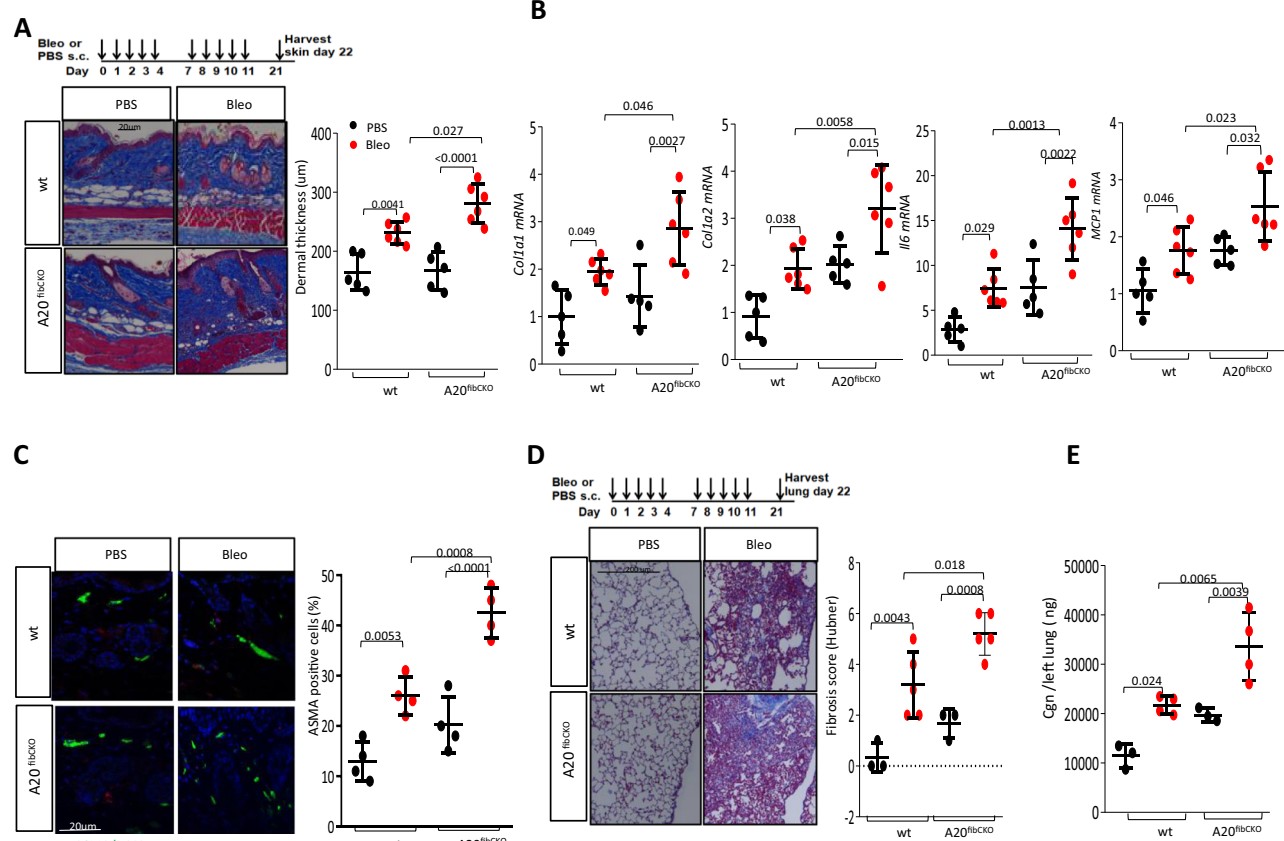

**Fig. 4 | Mice with fibroblast-specific A20 ablation show exaggerated skin and lung fibrosis.** A20fl/fl (wt) mice and A20fibcko mice were in parallel treated with s.c. bleomycin (5 mg/kg) or PBS for 2 weeks (5 days/week), and lesional skin and lungs were harvested on day 22 for analysis. **A**–**C** Skin fibrosis. **A** Top: schematic representation of the experiment. Left panels: Trichrome stain. Representative images. Bar = 20 μm. Right panel: dermal thickness (mean ± SD of 5 determinations/hpf); $n = 5$ (PBS-treated), $n = 6$ (Bleo-treated). **B** qPCR. Results, normalized with GAPDH, are mean ± SD of triplicate determinations. $n = 5$ (PBS-treated), $n = 6$ (Bleo-treated). One-way ANOVA followed by Sidak's multiple comparison test. **C** Left panel: immunolabeling with antibodies to ASMA. Scale bars = 20 μm. Right panel: proportion of immunopositive interstitial spindle-shaped fibroblasts within the dermis determined at four randomly selected locations/hpf. Results are mean ± SD ($n = 4$). One-way ANOVA followed by Sidak's multiple comparison test. **D**, **E** Lung fibrosis. **D** Top: schematic of the experiment. Left panel: Trichrome stain. Representative images; size bars = 200 μm. Right panel: lung fibrosis was determined (Hubner score); results are mean ± SD One-way ANOVA followed by Sidak's multiple comparison test. **E** Lung collagen content: each dot represents the mean ± SD of duplicate determinations ($n = 4$). One-way ANOVA followed by Sidak's multiple comparison test.

(Supplementary Fig. 10A). Moreover, the numbers of myofibroblasts, interstitial fibroblasts, and macrophages in the dermis were all significantly reduced ($p < 0.05$) in bleomycin-treated DREAM KO mice (Fig. 6E). Lungs from bleomycin-treated DREAM-null mice also showed significant attenuation of fibrosis and inflammation compared to lungs from wild-type control mice (Fig. 7 and Supplementary Fig. 10B), accompanied by elevated A20 expression (Supplementary Fig. 10A). Skin and lung fibroblasts explanted from DREAM-null mice showed attenuation of fibrotic responses when challenged by TGF-β ex vivo, with reduced stimulation of Type I collagen and ASMA expression, Smad2 activation, and markedly attenuated or absent upregulation of profibrotic genes (Fig. 7F and Supplementary Figs. 10B and 11). Notably, while TGF-ß treatment induced sustained downregulation of A20 in wild-type mouse fibroblasts, an entirely opposite effect was noted in DREAM-null fibroblasts treated in parallel with TGF-β (Fig. 7F and Supplementary Fig. 11). Hence, DREAM deficiency was accompanied by upregulation of endogenous A20, which might provide a mechanistic explanation for attenuated fibrotic responses in DREAM KO mice and DREAM-null fibroblasts.

## Targeting cellular A20 to mitigate fibrosis
In view of the previously unsuspected anti-fibrotic potential of A20 uncovered by these experiments, we posited that pharmacological augmentation of endogenous A20 might represent a novel therapeutic strategy to restrain fibroblast activation and limit fibrosis. We had previously shown that adiponectin, a 244-amino acid adipokine hormone whose expression in stromal cells is positively regulated by PPAR-γ[38,39], stimulated cellular A20 expression and exerted potent anti-fibrotic effects in vitro and in vivo[19]. Simulation of A20 was mediated through the adiponectin receptors AdipoR1 and AdipoR2[38]. To test the hypothesis that pharmacological targeting of adiponectin receptors might positively modulate A20 expression, we used AdipoRon, an orally active small molecule that binds to and activates AdipoR1 and AdipoR2 and downstream signaling pathways in a variety of cell types[40,41]. Chronic AdipoRon treatment mitigated bleomycin-induced fibrosis in wild-type mice[42]. Treatment of transiently transfected wild-type fibroblasts with AdipoRon induced a significant increase in A20-luc promoter activity (Fig. 8A, B). Stimulation of A20 transcription was seen as early as 60 min and persisted beyond 24 h. In contrast, a mutant A20-luc promoter lacking both NF-κB binding sites failed to respond to AdipoRon stimulation. Intriguingly, while in normal MEFs AdipoRon treatment induced significant stimulation of A20 (Fig. 8C) with concomitant suppression of type I collagen, in A20-null MEFs, AdipoRon failed to elicit anti-fibrotic activity (Fig. 8D). AdipoRon treatment of wild-type mice for four weeks resulted in prevention and

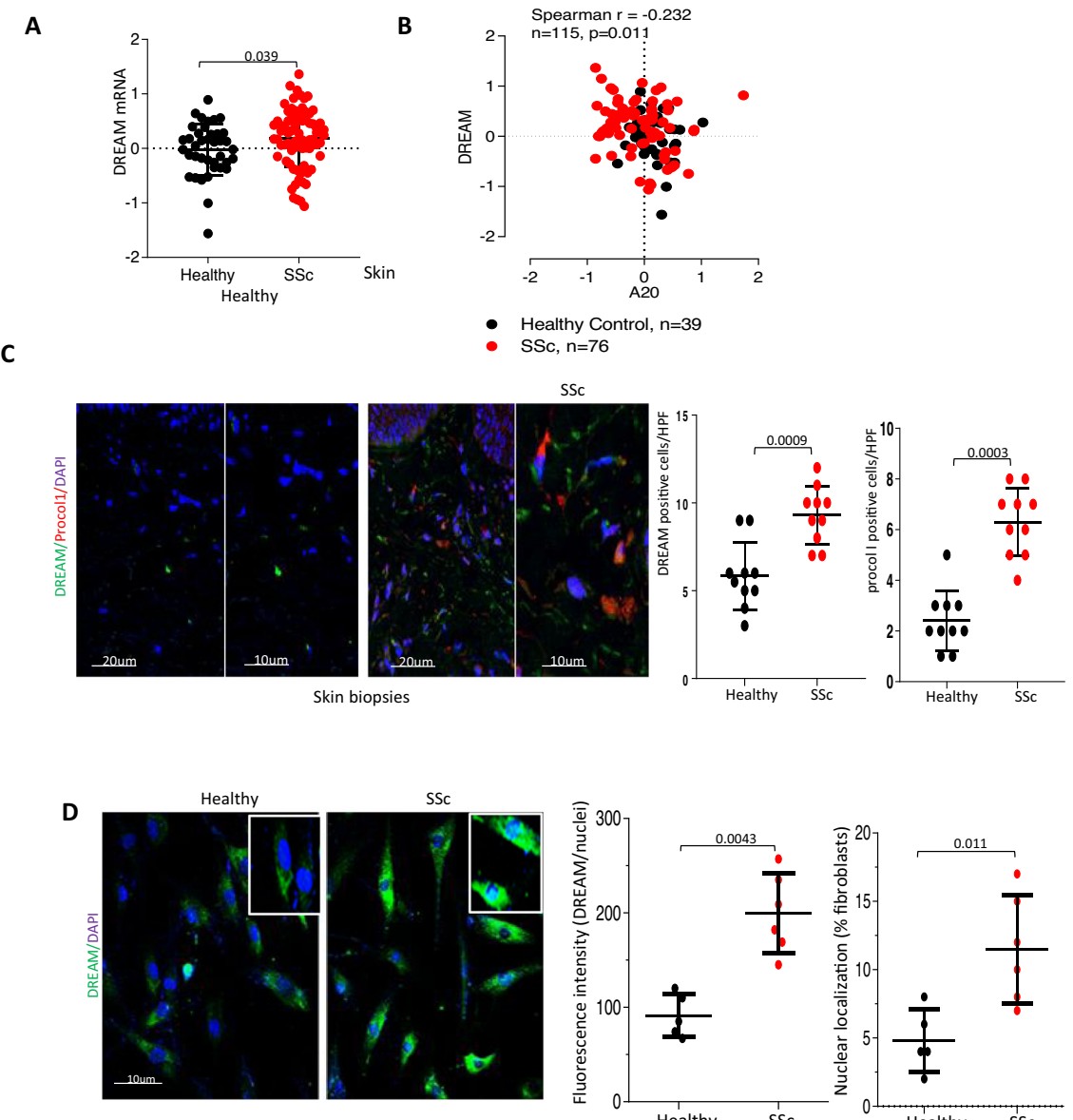

**Fig. 5 | Dream is elevated in SSc biopsies, and negatively correlated with A20.**
**A** DREAM mRNA expression in SSc skin biopsy-derived microarray datasets
(GSE59785, GSE45485, and GSE32413). Each dot represents a single subject (HC,
$n = 39$, SSc, $n = 76$). Two-tailed Mann–Whitney $U$ test. **B** Correlation of A20 and
DREAM mRNA in each biopsy. Spearman's correlation. **C** Immunolabeling of SSc
($n = 10$) and healthy control ($n = 10$) skin biopsies with antibodies to DREAM
and procollagen 1. Left panels: representative immunofluorescence images. Bars = 20
and 10 μm. Right panel: number of immunopositive cells determined from 4 hpf/

section in each biopsy. Two-tailed Mann–Whitney $U$ test. **D** Skin fibroblasts from
healthy control ($n = 5$) and SSc skin biopsies ($n = 6$) were immunolabelled using
antibodies against DREAM. Left panels: representative immunofluorescence ima-
ges (inset, high magnification). Bar = 10 μm. Middle panel: quantitation of DREAM
fluorescence (each point represents mean immunofluorescence intensity from four
randomly selected hpf). Right panel: percent of fibroblasts with predominantly
nuclear-localized DREAM (each point represents mean of 20 fibroblasts from each
cell). Two-tailed Mann–Whitney $U$ test.

---

reversal of fibrosis induced by bleomycin[43] that was accompanied by
upregulation of A20 in the lesional skin (Fig. 8E).

## Discussion

Genome-wide association studies have prominently linked various
*TNFAIP3* locus SNPs with autoimmune and inflammatory diseases
including psoriasis, rheumatoid arthritis, systemic lupus erythemato-
sus, Behcet's disease, and Crohn's disease[7]. Dysregulated A20 immu-
noregulatory functions that result in impaired control of NF-κB
signaling in conventional immune cells are implicated as a pathogenic
mechanism in these autoimmune and inflammatory conditions[7]. Gene
variants of A20 are also consistently linked with fibrotic SSc[5]. In con-
trast to inflammatory and autoimmune rheumatic conditions, the

clinical picture of SSc is dominated by fibrosis in multiple organs that
are attributed to the accumulation of activated tissue-resident
mesenchymal cells. The mechanistic links between altered A20 func-
tion and fibrosis in this context are completely unknown. Moreover,
while tissue-resident stromal cells show inducible A20 expression
comparable to conventional immune cells, there remains a limited
understanding of the regulation and function of A20 in stromal cell
types. We had demonstrated previously that A20 is expressed in
fibroblasts, and its overexpression inhibits profibrotic cellular
responses elicited by TGF-β[44]. In addition, A20 has also been shown to
negatively regulate signaling by Wnts, another family of profibrotic
stimuli implicated in SSc and other fibrotic conditions[7,19,20,45–47]. In light
of these observations suggesting a novel function for A20 in the

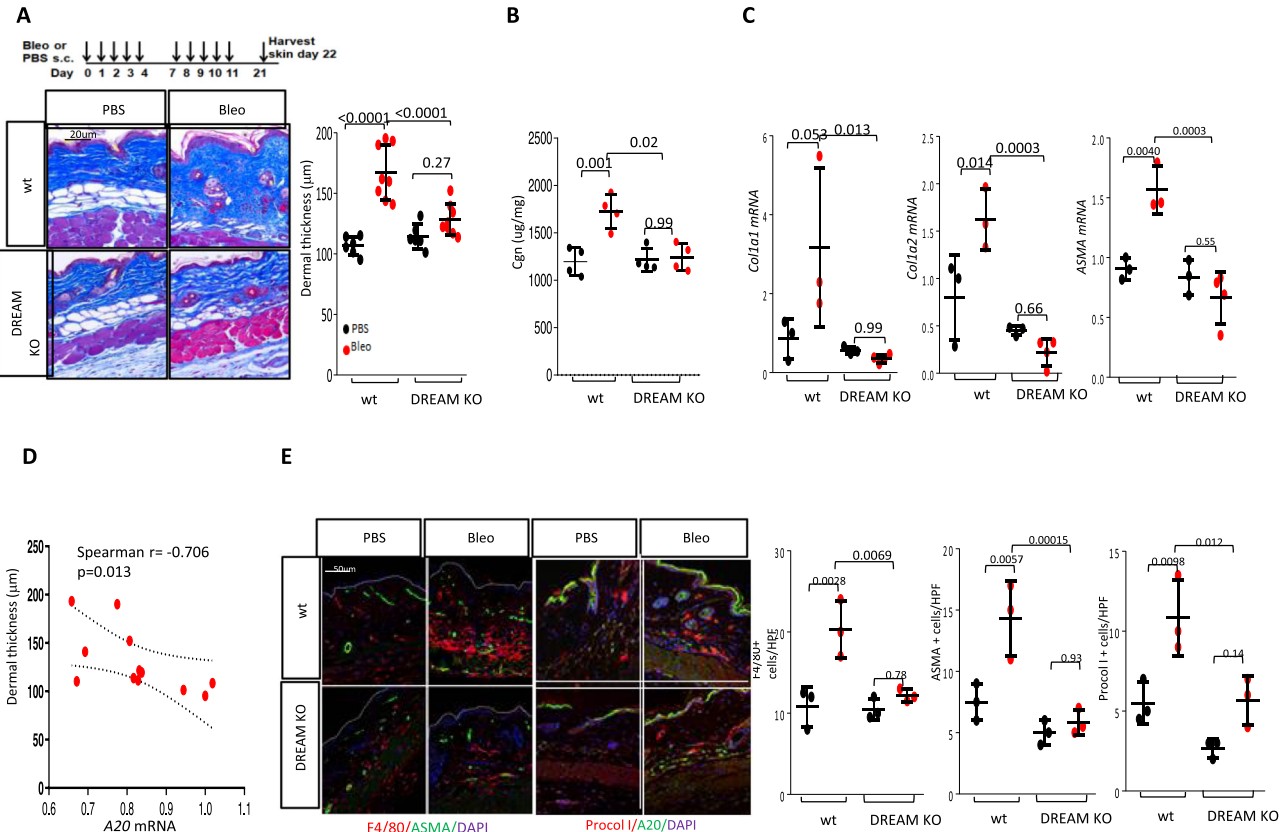

**Fig. 6 | Dream$^{-/-}$ mice showed attenuated skin fibrosis.** Eight-week-old C57/BL6 wild-type mice and DREAM$^{-/-}$ mice were treated in parallel with daily s.c. bleomycin (10 mg/kg/d) or PBS for 2 weeks, and lesional skin was harvested on day 22 for analysis. **A** Top: schematic of the experiment. Left panels: trichrome stain, representative images. Scale bar: 20 μm. Right panel: dermal thickness (mean ± SD of 5 determinations/hpf, PBS-treated, $n = 6$, Bleo-treated, $n = 8$). **B** Collagen content. One-way ANOVA followed by Sidak's multiple comparison test. **C** qPCR. Results, normalized with GAPDH, are mean ± SD of triplicate determinations from 3 mice per group. One-way ANOVA followed by Sidak's multiple comparison test. **D** Correlation between skin A20 mRNA levels and dermal thickness. Two-tailed Spearman's correlation. **E** Double immunolabeling with antibodies to ASMA/F4/80 and procollagen I/A20. Left panels: representative images; scale bars = 50 μm. Right panels: number of immunopositive cells within the dermis determined at four randomly selected locations/hpf. Results are mean ± SD from three mice/group. One-way ANOVA followed by Sidak's multiple comparison test.

modulation of fibrotic responses in stromal cells, here we sought to explore the expression of A20, its regulation and function in fibroblasts, and its potential pathogenic role in human SSc and preclinical disease models.

We demonstrate that A20 expression was reduced in SSc skin and lung biopsies, while expression of DREAM, the endogenous A20 suppressor, was elevated in these tissues. Mice genetically engineered for hypomorphic A20 expression or A20 deletion in a fibroblast-restricted manner demonstrated exaggerated fibrosis of skin and lungs, while in mice lacking DREAM, elevated endogenous A20 expression was accompanied by protection from fibrosis. In view of the consistent association of TNFAIP3 genetic variants with autoimmune and inflammatory conditions, the role and mechanism of A20 in their pathogenesis has been investigated in a variety of murine models. Global A20 deficiency in mice causes widespread tissue inflammation and perinatal lethality[15]. Conditional A20 deletion from hematopoietic lineages results in distinct autoimmune and autoinflammatory phenotypes[7]. To elucidate the mechanisms potentially linking A20 dysregulation and SSc pathogenesis, we generated A20 haploinsufficient mice that show a partial reduction of A20, comparable to the reduced levels of A20 in patients with SSc. At a young age, A20$^{+/-}$ mice showed no overt phenotype, but demonstrated spontaneous autoantibody production and increased fibrosis susceptibility. However, there was no further increase in autoantibody production with bleomycin treatment in A20$^{+/-}$ mice. It was previously shown that

mice hypomorphic for A20 in B cells spontaneously developed autoantibodies, suggesting cell-intrinsic functions of A20 in regulating B cell activation[48]. KEGG analysis of differentially expressed genes in the skin from A20$^{+/-}$ mice revealed significant enrichment of profibrotic pathways, including Wnt, that is consistent with the recent demonstration that A20 restricts β-catenin signaling in intestinal epithelial cells[45]. We confirmed elevated ß-catenin expression in dermal cells from bleomycin-treated A20$^{+/-}$ mice. Additional pathways enriched in A20$^{+/-}$ mice included FAK and TLR signaling. Moreover, we found increased TRAF6 ubiquitination in TGF-β-treated fibroblasts lacking A20 and activation of FAK and TLR signaling in A20$^{+/-}$ mice. These and other results implicate distinct profibrotic pathways involving TGF-ß, Wnt, and TLR, as well as FAK and TRAF6, as potential molecular targets of A20 in fibroblasts contributing to exaggerated fibrosis in the absence of A20. The parallels between skin fibrosis in A20$^{+/-}$ mice and in patients with SSc are underscored by a remarkable degree of gene regulation congruency between mouse and SSc, with 50% of genes sharing deregulated expression in both. Of potential relevance, it was recently reported that some patients with heterozygous loss of function mutations in the TNFAIP3 develop fibrosis in the liver[49].

Inflammation and dysregulated immunity are hallmarks of early-stage SSc, with an accumulation of monocytes, macrophages, and other innate immune cells: CD4 and CD8 T cells and B cells. These early immune responses are implicated in fueling subsequent fibrosis[50]. Despite compelling genetic association of A20 with fibrotic

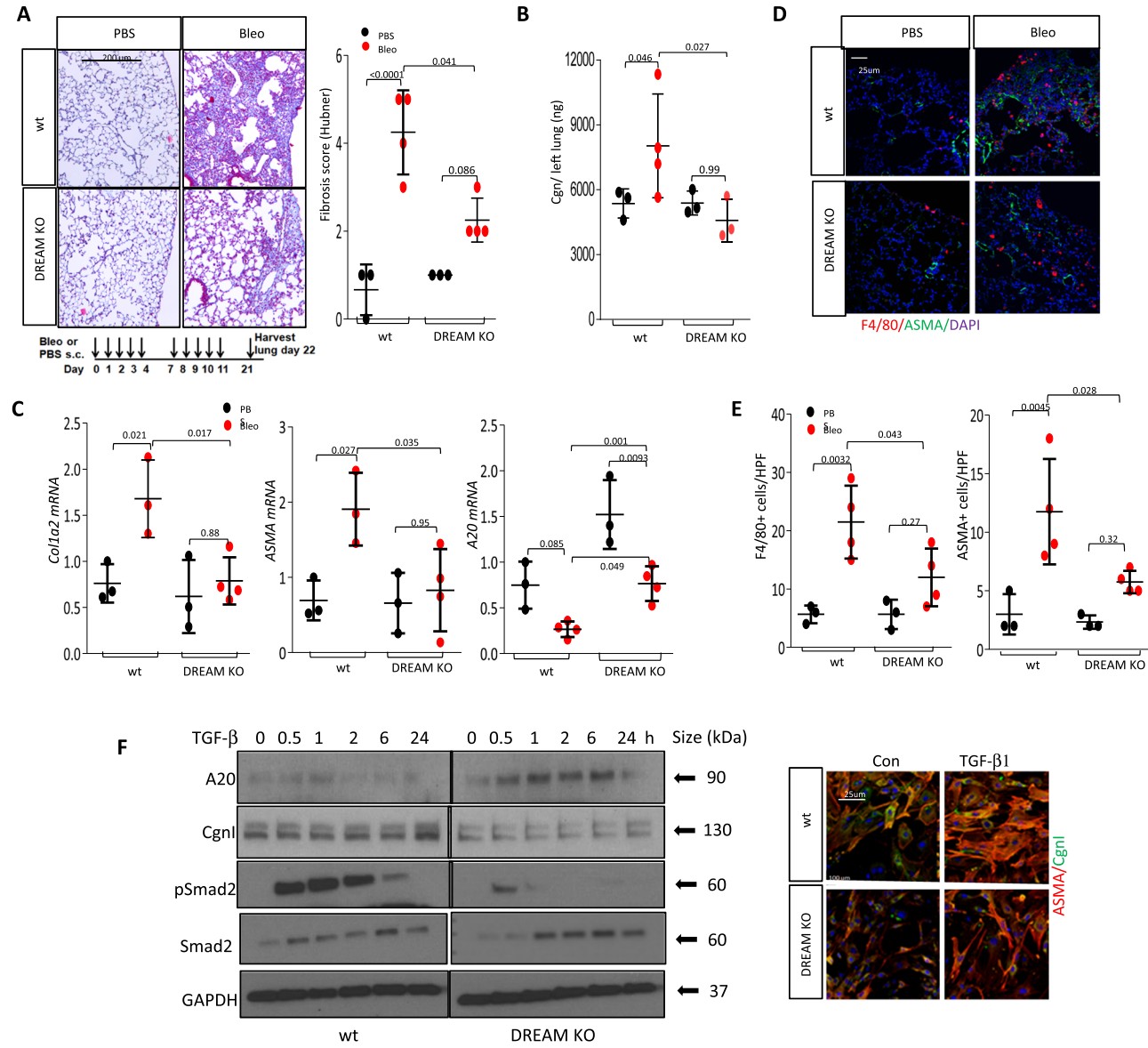

**Fig. 7 | DREAM$^{-/-}$ mice showed attenuated lung fibrosis.** Eight-week-old C57/BL6 wild-type mice and DREAM$^{-/-}$ mice in parallel were administered s.c. bleomycin or PBS for 2 weeks (5 days/week) and lungs were harvested on day 22. **A** Bottom: schematic representation of the experiment. Left panels: trichrome stain. Representative images; size bars = 200 μm. Right panel: lung fibrosis (Hubner score) in 8 hpf/mice. Results are mean ± SD from 4 mice (Bleo-treated) and 3 mice (PBS-treated) per group. One-way ANOVA followed by Sidak's multiple comparison test. **B** Collagen content. Results represent the mean ± SD from 3 (PBS-treated) or 4 mice (Bleo-treated) per group. One-way ANOVA followed by Sidak's multiple comparison test. **C** qPCR. Results, normalized with GAPDH, are mean ± SD of triplicate determinations from 3 (PBS-treated) or 4 mice (Bleo-treated) per group. One-way ANOVA followed by Sidak's multiple comparison test. **D** Immunolabeling with antibodies ASMA or F4/80. Scale bars = 25 μm. **E** Number of positive cells determined at four randomly selected locations/hpf. Results are mean ± SD from 3 (PBS-treated) or 4 mice (Bleo-treated) per group. One-way ANOVA followed by Sidak's multiple comparison test. **F** Lung fibroblasts explanted from DREAM$^{-/-}$ and wild-type mice were incubated in parallel with TGF-β for indicated time points. Left panels: whole-cell lysates examined by western analysis ($n = 2$). The samples derived from the same experiment and blots were processed in parallel. Right panels: immunofluorescence microscopy using antibodies to type I collagen (Cgn I) or ASMA ($n = 2$). Representative images ($n = 2$), bars = 25 μm.

SSc, the regulation and roles of A20 in fibroblast survival, differentiation, activation, and function have not been previously reported. Intriguingly, our results revealed reduced A20 expression in SSc skin and lung fibroblasts. Furthermore, mice with fibroblast-specific deletion of A20 showed enhanced sensitivity to fibrosis, suggesting a previously unrecognized vital role of A20 in mesenchymal cells to modulate their profibrotic response intensity and duration. The precise molecular mechanisms of how A20 regulates fibrosis-relevant intracellular signaling pathways in fibroblasts merit investigation.

Further experiments sought to delineate the mechanisms underlying reduced A20 expression in SSc. In SSc patients, reduced A20/ TNFAIP3 mRNA expression was associated with the rs117480515 A risk allele[51]. In the present studies, we found no evidence to support epigenetic mechanisms regulating A20 in SSc, and no significant difference in chromatin accessibility at the A20 locus, in healthy control and dcSSc fibroblasts[52]. While these observations suggest that genetic variants might contribute to reduced A20 expression in SSc, rather than epigenetic regulation, further work to investigate A20 downregulation in SSc is warranted. The multifunctional Ca$^{2+}$-binding protein DREAM has been studied for its role in pain and was recently shown to bind to DRE in the A20 promoter to suppress A20 expression[33]. However, DREAM has not been previously linked to fibrosis, and its expression, regulation, and function in SSc are

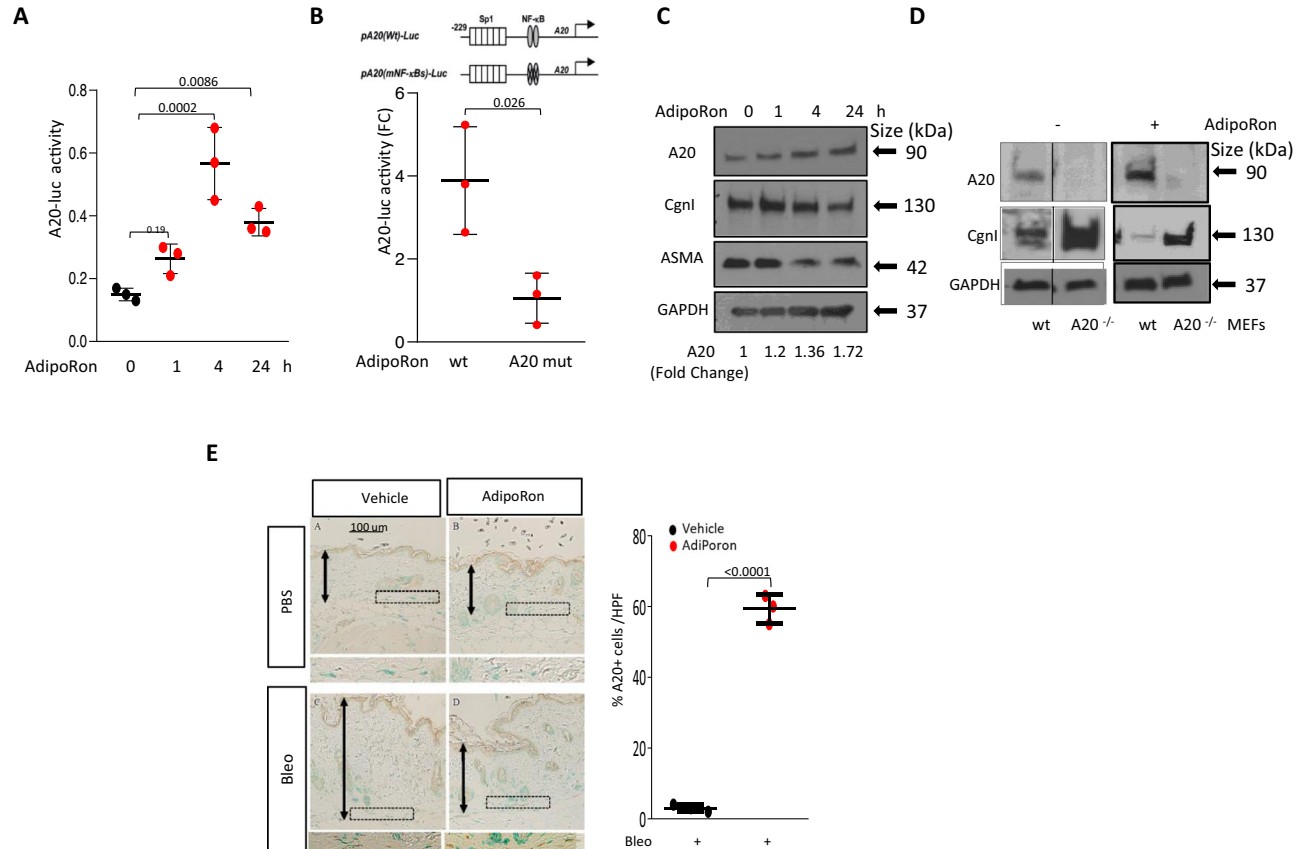

**Fig. 8 | Adiponectin receptor agonist adiporon induces a20 expression in explanted fibroblasts and in mice. A, B** Normal skin fibroblasts transiently transfected with A20-luc (**A**) or an NF-κB binding site mutated A20 (**B**) promoter-luc were incubated with AdipoRon (10 μg/ml) for indicated periods (**A**) or for 4 h (**B**), and whole-cell lysates were assayed for their luciferase activities. Results, normalized with Renilla luciferase ($n = 3$), shown as mean ± SD of triplicate determinations. **C** Left panels: Fibroblasts were incubated with AdipoRon (10 μg/ml) for indicated periods, and whole-cell lysates were examined by immunoblotting. Representative immunoblots ($n = 2$). A20 levels, normalized for GAPDH, are shown below each lane. Right panels: mouse embryonic fibroblasts from wild-type and A20[-/-] mice were treated with AdipoRon (10 μg/ml), and whole-cell lysates were examined by immunoblotting ($n = 2$). **D** C57Bl/6 mice were treated with bleomycin (200 μg/mouse) or PBS along with oral AdipoRon (50 mg/kg) given daily mice were sacrificed at day 22, and lesional skin was harvested for immunolabeling with antibodies to A20 ($n = 3$). **E** Left panels: representative images. Scale bar = 100 μm. Right panel: proportion of immunopositive interstitial cells within the dermis determined at four randomly selected locations/hpf. Results are mean ± SD ($n = 3$). One-way ANOVA followed by Sidak's multiple comparison test.

unknown. Here we found that DREAM was significantly upregulated in SSc skin biopsies and in explanted SSc skin fibroblasts, and its expression in skin biopsies showed anti-correlation with A20 levels. Pharmacological inhibition of histone acetylation reduced the levels of DREAM in explanted SSc fibroblasts, suggesting epigenetic mechanisms accounting for cell-autonomous dysregulation of A20 in SSc. We speculate that reduced expression of A20 in SSc might result from its transcriptional repression by DREAM. Indeed, consistent with this paradigm, we observed upregulation of A20 in multiple tissues from DREAM-null mice. Augmented A20 expression in these mice was associated with protection from bleomycin-induced fibrosis, revealing an entirely novel role for DREAM in modulating fibrosis propensity via endogenous A20. The importance of cellular DREAM in regulating A20 expression was further confirmed by DREAM knockdown in SSc fibroblasts, which resulted in augmented A20 and reduced collagen levels and myofibroblast markers. Although previously recognized primarily for its role in pain and inflammation[32], we now uncover a novel role of DREAM in promoting fibrosis. Indeed, in normal fibroblasts in culture, cellular DREAM was required for TGF-ß induction of full fibrotic responses.

In conventional immune cell types, expression of A20 is induced by inflammatory signals via the NF-κB pathway[53]. We and others have shown that adiponectin also induced A20 expression and persistence in both fibroblasts and macrophages via distinct mechanisms involving AMP kinase activation and PPARα[44,54]. We now show that the adiponectin receptor agonist AdipoRon directly stimulated cellular A20 expression, which may account for its previously reported anti-fibrotic activity[43]. Indeed, AdipoRon failed to reduce collagen production in A20-null MEFs, highlighting the essential role of inducible A20 in mediating this anti-fibrotic response. Of note, we found that A20 induction elicited by AdipoRon treatment appeared to be more durable and hence potentially more important, than A20 stimulation in these cells induced by LPS or TNF treatment.

In addition to AdipoRon, Vitamin E and various phytochemicals have been shown to exert immunomodulatory effects through A20 induction[34,55]. However, to our knowledge, none of these agents have been deployed in the clinic specifically for their ability to induce endogenous A20. Intriguingly, methotrexate was shown to exert its immunomodulatory effect in part via inducing endogenous A20 expression in leukocytes[56]. Moreover, MTX-responsive RA patients showed sustained elevation of A20 expression with no adverse side effects. Importantly, we and others show that DREAM[-/-] mice that show chronically elevated A20 expression, showed no overt phenotype[32,33]. Together these observations suggest that pharmacological induction of endogenous A20 may be well tolerated and not associated with significant immuno-compromise in mice; however, the safety and

tolerability of inducing A20 to modulate inflammation or fibrosis can only be determined by careful clinical observation.

Taken together, these results highlight a previously unrecognized mechanistic link between the A20-DREAM regulatory network and fibrotic disease manifestations of SSc and demonstrate a novel cell-autonomous regulatory role of A20 in mesenchymal cells that is distinct from its well-documented function in conventional immune cells. In both cellular and in vivo models, A20 had an inhibitory impact on fibrosis, while its transcriptional repressor DREAM was required for fibrosis. We propose that the upregulation of DREAM in SSc fibroblasts underlies suppression of A20, which in turn contributes to unchecked profibrotic signaling in stimulated fibroblasts. These cellular mechanisms might be operative in patients with SSc and contribute to disease pathogenesis. Accordingly, pharmacological approaches to augment endogenous A20 expression or function by blocking DREAM or by directly stimulating A20 expression via adiponectin receptor agonism might represent clinically viable therapeutic approaches in SSc and other intractable fibrosing conditions.

## Methods

### Human subjects

Subjects with SSc, SSc-ILD, and healthy volunteers were recruited from the Northwestern Scleroderma Program and the University of Pittsburgh School of Medicine. All patients fulfilled ACR criteria for the classification of SSc. Forearm skin biopsies were performed following obtaining written informed consent and in accordance with protocols approved by the Institutional Review Board for Human Studies at Northwestern University and lung transplant biopsies from the University of Pittsburgh School of Medicine. Lung tissues were obtained from SSc patients undergoing lung transplantation at the University of Pittsburgh Medical Center. Normal lung tissues were obtained from organ donors whose lungs were not used for transplant surgery. Demographics of the study subjects are shown in Table 1. Information obtained at the time of sample collection included subject demographics, disease duration (the interval from first non-Raynaud SSc manifestation), MRSS (range 0–51), and serum autoantibody status. Patients were subclassified into early (<24 months) or late-stage (>24 months) disease subsets. Written informed consent from the patients was obtained.

### Determination of A20 and DREAM levels

**Analysis of gene expression.** Levels of A20 and DREAM mRNA in forearm skin biopsies from SSc patients ($n = 72$) and healthy controls ($n = 39$) were determined in combined gene expression datasets[10] (NCBI GEO database under accession numbers GSE9285, GSE32413, GSE45485, and GSE59785).

**A20 and DREAM immunohistology.** A20 and DREAM expression was determined by immunolabeling of these proteins in skin (and lung) biopsies from 24 SSc patients and nine healthy adults (Cohorts 1 and 2). Paraffin-embedded skin and lung sections (4 μm thick) were deparaffinized and subjected to rehydration using series of gradient alcohol. Sections were then allowed for antigen unmasking using commercially available Dako retrieval buffer (Agilent Technologies, USA) for 30 min operated in steamer at 95 °C (Thermo Fisher Scientific), followed by cooling for 30 min. Sections were preincubated with 5% donkey serum blocking solution for 2 h at room temperature prior to incubation with primary rabbit antibodies against A20 (Abcam, ab92324; EPR2663, rabbit monoclonal, 1:200) or DREAM (Santa Cruz, sc166916, mouse monoclonal, 1:100) for overnight. Slides were washed three times with 1X phosphate-buffered saline, 0.1% Triton X-100 detergent (PBST) followed by incubation with mouse Alexa-fluor secondary antibodies (1:200) (Chicken anti-Rabbit 488 (Thermo Fisher, A21441, Lot:1796684); Chicken anti-Mouse 488 (Thermo Fisher, A21200, Lot:1078785) and 4,6-diamidino-2-phenylindone (DAPI) incubation for

nuclei staining, mounted with prolong glass antifade mounting media (Invitrogen)[57]. Activated fibroblasts were identified by immunolabeling using antibodies to procollagen 1 (EMD Millipore, 1:100). For immunohistochemistry, sections of paraffin-embedded skin were immuno-labelled with primary rabbit antibodies against A20 (Cohort 2). Slides were mounted, and images from immunofluorescence and immuno-histochemistry were evaluated under either a Nikon A1R laser scanning confocal microscope or Leica DMC4500 microscope. Negative controls stained without primary antibodies were used to confirm immuno-staining specificity.

### Experimental model of skin and lung fibrosis

All animal studies were conducted in accordance with NIH guidelines for the care and use of laboratory animals and under protocols approved by the Northwestern University IACUC. Animals were allowed free access to food and water ad libitium and maintained with 12/12 h light and dark cycle, with temperatures of 65–75 °F and 50–60% of humidity conditions. To generate A20$^{+/-}$ mice, A20$^{fl/fl}$ mice, C57BL/6 background (from Dr Averil Ma, University of California, San Francisco) were crossed with EIIa-Cre mice, C57BL/6 background (from Northwestern transgenic core) generating mice with a single A20 allele and displaying ~50% decrease in levels of A20 in multiple tissues[14,58]. At 6–8 weeks of age, female A20$^{+/-}$ mice were randomized to start treatment with s.c. bleomycin (5 or 10 mg/kg/day) or PBS given for 10 days (5 days/week). Mice were then sacrificed on day 22 and sera and tissues were harvested for analysis.

To create mice with fibroblast-specific A20 deletion (A20$^{fibcko}$ mice), mice carrying tamoxifen-inducible Cre (Cre-ERT) under the control of a fibroblast-specific Col1a2 enhancer were crossed with mice homozygous for the floxed A20 allele COL1A2-Cre-ERT (The Jackson Laboratory, Bar Harbor, ME) to generate COL1A2-Cre$^{+/-}$; A20$^{flox/flox}$ homozygous mice. At 3 weeks of age, female COL1A2-Cre$^{+/-}$; A20$^{flox/flox}$ mice were injected with 20 μg tamoxifen (Sigma) or vehicle (corn oil) i.p. for 7 days. One week later, treatment with s.c. bleomycin (5 mg/kg/day) or PBS daily was initiated and continued for 10 days (5 days/week) (Supplementary Fig. 4). Mice were sacrificed on day 22, and lesional skin and lungs were harvested for analysis. In other experiments, bleomycin of PBS treatment of 8-week-old female DREAM$^{-/-}$ mice (gift from Dr Chinnaswamy Tiruppathi, UIC, Illinois) and littermate control was continued for 10 days, mice were sacrificed on day 22, and lesional skin and lungs were harvested for analysis. Each experiment was repeated at least two times with consistent results. In a complementary experiment, a non-inflammatory fibrosis model was used. Eight-week-old female Tsk1/+mice (C57BL/6 background, The Jackson Laboratory) were crossed with A20$^{+/-}$ mice to generate A20$^{+/-}$; Tsk1/+mice. At 12 weeks of age, A20$^{+/-}$ Tsk1/+ mice and Tsk1/+ mice were sacrificed, and lesional skin was harvested for analysis.

Paraffin-embedded tissue sections (4 μm) were stained with hematoxylin and eosin, or Trichrome, and dermal thickness was determined by measuring the length of dermis (both papillary and reticular) excluding epidermis and adipose layer in all experimental groups and then compared the thickness amongst the groups[57]. Lung tissues were stained with H&E and scored for fibrosis in a blinded manner by an expert pulmonary pathologist (Anjana Yeldandi, Northwestern University)[57]. For immunofluorescence analyses, paraffin-embedded skin and lung sections were deparaffinized and subjected to rehydration using series of gradient alcohol. Sections were then allowed for antigen unmasking using commercially available Dako retrieval buffer (Agilent Technologies, USA) for 30 min operated in steamer at 95 °C (Thermo Fisher Scientific), followed by cooling for 30 min at room temperature. Sections were preincubated with 5% donkey serum blocking solution for 2 h at room temperature prior to incubation with primary rabbit antibodies specific against F4/80 (1:500, eBioscience, 14-4801-82, rat monoclonal, BM8 clone, San Diego, CA), αSMA (1:100, Abcam, ab5694, rabbit polyclonal),

procollagen I (1:100, EMD Millipore, MAB1912, MAB1912, M-58 clone), Perilipin (1:100, Abcam, ab61682, goat polyclonal), β-catenin (1:100, Abcam, ab32572, rabbit monoclonal, E247 clone), and A20 (1:100, Abcam, ab92324, rabbit monoclonal, EPR2663 clone) overnight, followed by washes with PBST and incubation with Alexa-fluor-labeled rabbit secondary antibodies (1:200) (Chicken anti-Rabbit 488 (Thermo Fisher, A21441, Lot:1796684); Chicken anti-Mouse 488 (Thermo Fisher, A21200, Lot:1078785); Chicken anti-Rat 594 (Thermo Fisher, A21471, Lot:1003225); Donkey anti-Goat 488 (Thermo Fisher, A11055, Lot:1687906); Goat anti-Rabbit 594 (Thermo Fisher, A11037, Lot: 1915919); goat anti-rabbit 647 (Thermo Fisher, A21245, lot: 1558736) on the next day for 1 h. Slides were washed with PBST and nuclei were detected using DAPI. Collagen content of the lungs was determined by hydroxyproline assays (Colorimetric Assay Kits, Biovision, Milpitas, CA). Total RNA isolated from mouse skin and lung tissues was reverse transcribed to cDNA using Supermix and analyzed by real-time qPCR (Applied Biosystems) on an Applied Biosystems 7500 Prism Sequence Detection System[57].

### Circulating autoantibody assays

Using autoantigen arrays where autoantigens are immobilized onto nitrocellulose-coated slides, arrays were hybridized with diluted serum samples from A20[fl/fl] and A20[+/−] mice (Autoantibody profiling service, UT Southwestern)[59]. The autoantibodies bound to their corresponding antigens on the array are detected with biotinylated secondary antibodies against IgG isotype and with fluorescent-labeled anti-biotin antibodies for imaging as described[59]. Heatmap of IgG autoantibodies after filtered by SNR >3% (considered significantly greater than background and true signals) across all samples is generated by Cluster and TreeView software.

### Cell culture and reagents

Fibroblasts were explanted from forearm skin biopsies from healthy controls and SSc patients or neonatal mouse skin and lung. Fibroblasts at low passage were cultured in monolayers and maintained in Dulbecco's Modified Eagles Medium (DMEM) (Thermo Fisher Scientific) supplemented with 10% fetal bovine serum and 1% antibiotic-antimycotic (Invitrogen Life Technologies) as previously reported[60]. Wild-type and A20[−/−] MEFs were cultured using the same media as described. For experiments, cultures were placed in serum-free media containing 0.1% BSA (Millipore Sigma) or low serum media (containing 0.1% FBS (Thermo Fisher Scientific) with indicated concentrations of TGF-ß1 (10 ng/ml) (PeproTech).

### Fibroblast lines identification

Foreskin fibroblasts were routinely prepared from anonymous discarded foreskins from circumcised young individuals at Feinberg Hospital Northwestern University using the standard explant method. SSc skin fibroblasts and fibroblasts from age-matched healthy control subjects were routinely generated from 3 mm punch biopsies from the forearm. Fibroblast lines were established by standard explant methods and propagated. Fibroblasts were stored in the Northwestern Scleroderma Biorepository. Wild-type and A20-null MEFs (mouse skin and lung) were obtained from the laboratory of Dr Averil Ma and tested for the absence of A20 using both A20-specific antibodies and RT-PCR. Mouse skin and lung fibroblasts were obtained by standard explant methods from DREAM-null mice; while mouse skin, lung, and spleen fibroblasts were obtained by standard explant method from fibroblast-specific inducible A20 ablation (A20[fibcko]).

### RNA isolation and analysis

Total RNA isolated from fibroblasts cultures and human and mouse skin biopsies was reverse transcribed to cDNA using Supermix (cDNA Synthesis Supermix; Quanta Biosciences) and analyzed by real-time qPCR[61,62]. The products (20 ng) were amplified using SYBR Green PCR Master Mix (Applied Biosystems) on an Applied Biosystems 7500 Prism Sequence Detection System. The results were normalized to GAPDH RNA levels, and fold change in samples was calculated as $2^{-\Delta\Delta Ct}$ ($2^{-((Ct\ target - Ct\ GAPDH)\ treatment\ -\ (Ct\ target - Ct\ GAPDH)\ non-treatment)}$) as described. In a separate experiment, dcSSc fibroblasts were treated with BET inhibitor JQ1 (1 µM) for 48 h before the cells were collected for mRNA extraction. After cDNA synthesis, the samples were subjected to qPCR using ViiA 7 Real-Time PCR System. Sequences of the primers are shown in Supplementary Table 2.

### Genome-wide expression profiling and data analysis

To examine the modulation of gene expression at the genome-wide level, RNA was isolated from skin biopsies from bleomycin-treated and untreated mice using RNeasy fibrous tissue minikits (Qiagen, Valencia, CA). The integrity of RNA was ascertained using an Agilent Bioanalyzer (Agilent Technologies, Santa Clara, CA), and cDNA was labeled using Ambion labeling kits (Ambion) and hybridized to Illumina human HT12 v4 Expression Microarray Chips (Illumina, San Diego, CA), as previously described[60]. Analysis of differential gene expression between groups was performed with DESeq2 R package, using a model based on the negative binomial distribution. To control for multiple testing and reduce the FDR, stringent statistical criteria were used to identify differentially expressed genes with raw $p < 0.01$ and FDR-adjusted $p < 0.05$. DESeq2 R package is used to generate the plots. KEGG enrichment analysis of differentially expressed genes was executed by R package cluster Profiler. KEGG terms with adjusted $p < 0.05$ were considered significant enrichment.

### Western and immunoprecipitation analyses

At the end of the experiments, fibroblasts were harvested. Culture supernatants were collected from the cell culture media and the whole-cell lysates were prepared and equal amounts of proteins (10–20 µg/lane) and media were subjected to SDS electrophoresis, followed by transfer of the samples onto PVDF (Immobilon, Millipore)/NC (Amersham Protran) membranes. The blots were incubated with 10% dry powder milk (Research products, International) for 1 h at room temperature to avoid non-specific binding. The membranes were probed using primary antibodies specific for human Type I collagen (Southern Biotechnology, 1310-01, Goat IgG), A20 (Santa Cruz, Sc-166692, mouse monoclonal IgG2a), α-smooth muscle-actin (Sigma, 5228, mouse monoclonal, AC-74), pSmad-2 (Cell Signaling Technology, 3108S, rabbit IgG), β-Actin (Sigma-Aldrich, A5441, mouse monoclonal, AC-15) and GAPDH (Santa Cruz, sc-365062, mouse monoclonal IgG1k) overnight. The following day, membranes were washed three times with 1X Tris-buffered Saline, 0.1% tween 20 detergent for 10 min for each wash followed by incubation with respective secondary antibodies and detection of the membrane using commercially procured super signal west pico plus chemiluminescent substrate (Thermo Fisher Scientific)[10]. In other experiments, whole-cell lysates prepared from TGF-β (10 ng/ml) treated confluent cultures of wild-type and A20[−/−] MEFs (1 mg) were immunoprecipitated with antibodies to P4D1 (Santa Cruz, Dallas, TX) followed by immunoblotting using antibodies to ubiquitin TRAF6 (Santa Cruz) and P4D1.

### Transient transfection assays

Subconfluent fibroblasts in culture were co-transfected with A20-luc, NF-κB mutated A20-luc, and Renilla luciferase pRL-TK plasmids (Promega) using Lipofectamine reagent (Thermo Fisher Scientific)[17]. Cultures were harvested following indicated incubation period, and whole-cell lysates normalized with pRL-TK were assayed for their luciferase activities The experiment was performed in triplicate and repeated twice with consistent results. In separate experiments, near-confluent SSc fibroblasts were transiently transfected with antisense-DREAM plasmid or control plasmid. Cells were fixed and immunolabelled.

## Immunofluorescence confocal cytochemistry using skin, lung fibroblasts, and mouse embryonic fibroblasts (MEFs)

Fibroblasts seeded on 8-well Lab-Tek II chamber glass slides (Nalgene Nunc International, Naperville, IL) were incubated in serum-free DMEM with or without TGF-β (10 ng/ml) for 24 h. Cells were then fixed with 4% paraformaldehyde (Thermo Fisher Scientific), permeabilized with 0.2% Triton X-100 (Thermo Fisher Scientific), and incubated with blocking solution (10% dry powder milk, Research products, International) followed by incubation with antibodies specific to α-smooth muscle-actin (αSMA, 1:200, abcam, ab5694, rabbit polyclonal), type I collagen (1:200, (Southern Biotechnology, 1310-01, Goat IgG), phospho-FAK (1:100, Cell Signaling, 3284T, rabbit polyclonal), phospho-TAK1 (1:100, Cell Signaling, 4531S, rabbit IgG monoclonal), FnEDA (1:200, Sigma, F6140, Mouse IgM Monoclonal, FN-3E2), A20 (1:200, Abcam, ab92324, rabbit monoclonal, EPR2663), and DREAM (1:100, Santa Cruz, sc166916, mouse monoclonal IgG2bk) overnight followed by Alexa-fluor-labeled secondary antibodies Chicken anti-Rabbit 488 (Thermo Fisher, A21441, Lot:1796684); Donkey anti-Goat 594 (Thermo Fisher, A11058, Lot:1445994); Donkey anti-Mouse 594 (Thermo Fisher, A21203, Lot:1163390); Chicken anti-Mouse 488 (Thermo Fisher, A21200, Lot:1078785) for 1 h at room temperature. Cells were washed thrice with PBST, incubated with DAPI (Sigma) for 10 min, and mounted with prolong glass antifade mounting media (Invitrogen)[37]. Nuclei were identified using DAPI, and immunofluorescence was evaluated under Nikon A1C confocal microscope.

## Statistical analysis

The sample size was estimated based on our previous experiments for both human and mouse studies. No statistical method was used to predetermine sample size as for both human and mouse studies enough samples were available at the same time. Experiments and quantifications for patients' study were not done in a blinded fashion as the samples were archived with specimen id as shown in Table 1 (patient demographic information). There were no exclusion criteria for both human and animal experiments. Mice were randomized into different groups. Explanted fibroblasts from healthy and patient samples were randomized. Studies using analysis of human data were replicated by analyzing data from independent validation cohort when available, as indicated in the results. Experiments with mice were replicated twice. Two-sided Mann–Whitney $U$ test and Student's $t$-test were used for comparisons between two groups, with a $p$ value < 0.05 considered statistically significant. Comparisons among three or more groups were performed using analysis of variance followed by Sidak's correction for multiple comparisons. Two-sided Spearman's rank correlations were calculated to measure the correlation between A20 and DREAM levels. Data are presented as mean ± SD unless otherwise indicated. Data were analyzed using GraphPad Prism (GraphPad Software version 8, GraphPad Software Inc., CA).

## Study approval

Human studies were approved by the IRBs of Northwestern University and the University of Pittsburgh. All participants provided written informed consent.

## Reporting summary

Further information on research design is available in the Nature Research Reporting Summary linked to this article.

## Data availability

Accession code for RNA sequencing submission is GSE194380. Source Data are provided with this paper.

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

## Acknowledgements

We are grateful to members of the Scleroderma Research Laboratory, the staff of the Mouse Histology & Phenotyping Laboratory (MHPL), and the Northwestern University Skin Diseases Research and Imaging Cores

for excellent technical support. We acknowledge the unreserved support from Dr Carol Feghali-Bostwick for providing lung biopsy specimens collected from SSc-ILD transplant biopsies and Jose Naranjo for providing an antisense-DREAM plasmid. The study was supported by grants from the National Institutes of Health, National Institute of Arthritis and Musculoskeletal and Skin Diseases (NIAMS AR42309), and the Scleroderma Foundation.

## Author contributions

S.B. and J.V. designed the experiments. W.W., D.X., S.B., S.Ba., K.A., H.S., T.Y., J.J., B.Y., P.S.T., and R.G.M. performed experiments. S.B., H.A.-V., S.Ba., S.D.M., B.C.S., A.Y., Y.A., S.S., D.B., S.D., H.A.-V., A.H.S., and P.S.T. analyzed the data. E.H., C.T., and B.C.S. provided intellectual input and feedback on the manuscript. S.B. and J.V. prepared the manuscript.

## Competing interests

The authors declare no competing interests.

## Additional information

**Wenxia Wang**[1], **Swarna Bale**[1,2], **Jun Wei**[1], **Bharath Yalavarthi**[2], **Dibyendu Bhattacharyya**[2], **Jing Jing Yan**[1], **Hiam Abdala-Valencia**[3], **Dan Xu**[4], **Hanshi Sun**[2], **Roberta G. Marangoni**[1], **Erica Herzog**[5], **Sergejs Berdnikovs**[6], **Stephen D. Miller**[4], **Amr H. Sawalha**[7], **Pei-Suen Tsou**[2], **Kentaro Awaji**[8], **Takashi Yamashita**[8], **Shinichi Sato**[8], **Yoshihide Asano**[8], **Chinnaswamy Tiruppathi**[9], **Anjana Yeldandi**[10], **Bettina C. Schock**[11], **Swati Bhattacharyya**[1,2] ✉ & **John Varga**[1,2] ✉

[1]Northwestern Scleroderma Program, Department of Medicine, Feinberg School of Medicine, Chicago, IL, USA. [2]Michigan Scleroderma Program, Department of Internal Medicine, University of Michigan, Ann Arbor, MI 48109, USA. [3]Division of Pulmonary and Critical Care Medicine, Department of Medicine, Feinberg School of Medicine, Northwestern University, Chicago, IL 60611, USA. [4]Department of Microbiology-Immunology, Northwestern University Feinberg School of Medicine, Chicago, IL 60611, USA. [5]Department of Medicine, Yale University School of Medicine, New Haven, CT, USA. [6]Division of Allergy and Immunology, Department of Medicine, Northwestern University Feinberg School of Medicine, Chicago, IL 60611, USA. [7]Department of Medicine, University of Pittsburgh School of Medicine, Pittsburgh, PA, USA. [8]Department of Dermatology, University of Tokyo Graduate School of Medicine, Bunkyo-ku, Tokyo 113-8655, Japan. [9]Department of Pharmacology and Center for Lung and Vascular Biology, College of Medicine, University of Illinois, Chicago, IL, USA. [10]Department of Pathology, Feinberg School of Medicine, Northwestern University, Chicago, IL 60611, USA. [11]Wellcome-Wolfson Institute for Experimental Medicine, Queens University Belfast, Belfast, UK. ✉e-mail: bhattasw@med.umich.edu; vargaj@med.umich.edu

