## [Peer Review File · Nature Communications]

Reviewers' Comments:

Reviewer #1:

Remarks to the Author:

In this manuscript, Wang et al. provide evidence that the expression of the ubiquitin-editing enzyme A20/TNFAIP3 is downregulated and associated with fibrosis in systemic sclerosis (SSc). Conversely, expression of DREAM, a transcriptional repressor of A20, is upregulated in SSc. The authors propose that the A20-DREAM regulatory network is a key regulator of the fibrotic process and pathogenesis in SSc. The overall approach is rigorous and includes data from SSc patients as well as animal fibrosis models with A20 haploinsufficient and fibroblast-specific knockout mice and DREAM-null mice. This study is for the most part convincing and conclusions are justified; however, it could be strengthened by addressing the mechanisms of A20 inhibition of fibrosis in the context of SSc.

- 1) What are the potential targets and pathways regulated by A20 that are relevant in SSc? Is it a TLR pathway (e.g., TLR4) promoting fibrosis and the development of SSc? TRAF6 is a well known substrate of A20 in TLR pathways. Are there any differences in TRAF6 protein expression or activation (i.e., ubiquitination) in fibrotic tissues from A20 or DREAM KO mice treated with bleomycin? Alternatively, is A20 regulation of the Wnt pathway important for the inhibition of fibrosis? Is beta-catenin protein expression increased in fibrotic tissues from A20 or DREAM KO mice treated with bleomycin?
- 2) The authors propose that DREAM suppresses A20 expression in the context of SSc. How do A20/TNFAIP3 SNPs associated with SSc factor into this model? Is the A20-DREAM regulatory network the primary determinant of fibrosis or are there other mechanisms of A20 downregulation in SSc (i.e., epigenetic or post-transcriptional)?
- 3) The IF staining of DREAM in healthy and SSc skin fibroblasts (Fig. 5D) indicate that DREAM is mainly localized in the cytoplasm. This result seems to contradict the authors' model that DREAM suppresses the transcription of the A20/TNFAIP3 gene in the nucleus.
- 4) In Fig. 7F, authors should include a blot for total Smad2 as a control. Why is Smad2 phosphorylation decreased in DREAM KO cells? Is this due to A20 overexpression? The basal levels of A20 appear similar in WT and DREAM KO cells. Authors should quantify A20 bands relative to GAPDH.
- 5) In Fig. 8C, AdipoRon does not appear to robustly induce A20 expression. How does AdipoRon compare with TNF or LPS treatment for the induction of A20 in fibroblasts?

Minor points:

- 1) It seems that the expression of A20 mRNA in SSc skin is just more variable compared to healthy controls (Fig. 1A).
- 2) Why is there no statistically significant difference in Col1a1 mRNA between WT and A20 fibroblast mice treated with bleomycin (Fig. 4B)?

Reviewer #2:

Remarks to the Author:

This is a well-designed and -written submission from one of the leading groups in systemic sclerosis. In this manuscript, the authors proposed that ubiquitin-editing enzyme A20 is a main contributor to the susceptibility of developing fibrosis in systemic sclerosis (SSc). The suggested mechanism of A20 in driving fibrosis depends on the levels of its negative transcriptional regulator, DREAM, in tissues of interest. When DREAM is present, the expression of A20 is reduced with an increase of phenotypical characteristics of SSc. The authors demonstrate that A20 expression is reduced in SSc tissue, while DREAM levels are elevated. A20 haplo-insufficient mice confirmed a

similar phenotype between the mouse and human SSc skin biopsies. The A20 levels affect survival, differentiation, and activation of fibroblasts, thereby increasing the sensitivity for development of fibrosis. DREAM's regulation of the A20 response uncovers a novel fundamental role for this transcriptional repressor, known primarily as a modulator of pain and inflammation. The authors propose this feedback network as a possible pharmacological target to augment A20 expression and reduce fibrosis in SSc.

Specific points:

- 1) The methods used for immunofluorescence quantification are not clear and should be explained in detail in material and methods.
- 2) Figure 8D is an important figure, because it shows the reversal of skin fibrosis. Why is the depicted H&E staining so different, when compared to the other H&E's presented in the paper? Format and labels are also different, they should be consistent with the other figures.
- 3) The authors suggest the use of pharmacological approaches to augment endogenous A20 as a therapeutic approach in SSc. What will be the effect of this intervention in normal tissues? What will be the effect of such treatment in the intensity and duration of other inflammatory responses? The authors should discuss the limitations and caveats of this approach.

Reviewer #3:

Remarks to the Author:

Wang et al. present a study about the contribution of A20 and DREAM to fibrosis in systemic sclerosis and thus continue previous studies. However, as presented, the study is not convincing and I have major concerns to improve the manuscript.

Comments:

1. Figure 1B: The quality / resolution of the IF pictures should be improved, also the scale bar should be better visible. Quantitation: The authors measured fluorescence intensity, but how were fibroblasts identified, as they give % of positive fibroblasts? In addition, how were unspecific signals identified?
2. Figure 1C: What is the exact size of the bar? The legend says 10 μm , the writing in the pictures say 20 μm and 5 μm . This discrepancy can also be found in Fig. 1E (20 μm vs. 50 μm).
3. Figure 2A: Instead of labelling the fold change as low and high, it would be better to give fold change values corresponding to the different color shades.
4. Figure 2B: Why were only three mice per group analyzed by qPCR for Col1a1 and Col1a2, and not the whole groups?
5. Figure 2E: The authors used αSMA for identification of activated fibroblasts. However, αSMA is also highly expressed in vessels. How did the authors distinguish fibroblasts and vessels, especially small vessels? The standard deviations in the quantitations are in part quite high. More samples could solve this problem.
6. Figure 2F: The right trichrome-stained picture (A20+/- Tsk1/+) looks a bit squeezed.
7. Figure 3: In panel A, it would be better to show low magnification of the whole lung section (at best) and then magnify a certain part.
How many samples have been analyzed for each read-out?
Panels D and E: Same comment as for Fig. 2E: How were fibroblasts identified and discriminated from vessels?
8. Figure 4: Panel B: In contrast to the statement in the text, there are no significant increases for Col1a1 (for all comparisons). In addition, how do the authors explain that there is no increase in wildtype mice injected with bleomycin (the only exception is IL6)? There should be a significant induction at least of collagen expression also in wildtype mice injected with bleomycin. Panel C: According to the dots in the graph, the authors analyzed with an N of 2 and 3? This is statistically not convincing. For the immunofluorescence staining, again a cell type marker to discriminate αSMA -positive fibroblasts from vessels is missing.
9. Figure 5, panel C: There is again a discrepancy about the bar size: 50 μm in the legend, 20 μm or 10 μm in the picture.
10. Figure 6. Panel B and C: Again n = 3 with high standard deviations are statistically not

convincing. Panel E, and Fig. 7D: Same as for previous figures and stainings, how were fibroblasts identified and discriminated from vessels?

11. In general, the sample numbers in many read-outs are very low with $n = 3$ or even $n = 2$.

12. The mechanism of DREAM upregulation in SSc is missing and should be analyzed.

13. To proof whether DREAM is suppressing A20 by direct binding to the promoter in SSc can be easily evaluated by chromatin immunoprecipitation in SSc fibroblasts.

14. The authors revealed autoantibodies in the A20 +/- mice. Did they investigate the contribution of B cells and other leukocytes to the higher susceptibility of these mice to fibrotic stimuli?

15. In the methods section, the authors should explain how they analyzed the genome-wide expression profiling in more detail. Which software and packages were used? Did they only use one software/package or did they confirm with at least a second one? How did they generate the plots?

REVIEWER COMMENTS

Reviewer #1 (Remarks to the Author):

In this manuscript, Wang et al. provide evidence that the expression of the ubiquitin-editing enzyme A20/TNFAIP3 is downregulated and associated with fibrosis in systemic sclerosis (SSc). Conversely, expression of DREAM, a transcriptional repressor of A20, is upregulated in SSc. The authors propose that the A20-DREAM regulatory network is a key regulator of the fibrotic process and pathogenesis in SSc. The overall approach is rigorous and includes data from SSc patients as well as animal fibrosis models with A20 haploinsufficient and fibroblast-specific knockout mice and DREAM-null mice. This study is for the most part convincing and conclusions are justified; however, it could be strengthened by addressing the mechanisms of A20 inhibition of fibrosis in the context of SSc.

1) What are the potential targets and pathways regulated by A20 that are relevant in SSc? Is it a TLR pathway (e.g., TLR4) promoting fibrosis and the development of SSc? TRAF6 is a well known substrate of A20 in TLR pathways. Are there any differences in TRAF6 protein expression or activation (i.e., ubiquitination) in fibrotic tissues from A20 or DREAM KO mice treated with bleomycin? Alternatively, is A20 regulation of the Wnt pathway important for the inhibition of fibrosis? Is beta-catenin protein expression increased in fibrotic tissues from A20 or DREAM KO mice treated with bleomycin?

Responses: We thank the Reviewer for raising this important issue. We speculate that multiple SSc-relevant pathways might be targeted by A20. To address this issue, we performed a series of new *in vitro* and *in vivo* experiments and data analyses. Based on KEGG pathway analysis (page 9) of bleomycin-treated-A20^{fl/fl} mice transcriptome datasets (Table 3), we focused our attention on potential targets of A20 in SSc-relevant pathways.

These include:

1. Wnt, TLR and TGF- β signaling
2. FAK, β -catenin, TRAF6 and TAK1

We performed additional experiments and analysis in skin and lungs from A20^{fl/fl} and A20^{+/-} mice, and *in vitro* experiments with wildtype and A20-null

MEFs to confirm predictions from KEGG analysis.

We previously published (Bhattacharyya S. et al., Arthritis Res Ther. 2016) that A20 repressed TGF- β and TLR-mediated fibrotic and canonical TGF- β signaling. Additional new experiments showed that TGF- β in A20^{-/-} fibroblasts induced exaggerated FAK activation, implicating FAK in fibroblasts as a molecular target of A20.

In terms of Wnt/ β -catenin signaling, we found increased β -catenin in the skin and lung from

See pages 9, 10, 29, 30, 36, 37,38 and Table 3 and Suppl. Figs. 5 and 6

bleomycin-treated A20^{+/-} mice, identifying it as a potential A20 target.

Regarding TLR as an A20 target of the anti-fibrotic effects of A20, it was challenging to assess TRAF6 ubiquitination in human or mouse tissues due to abundance of non-specific Ub molecules. To get around this challenge, we therefore studied TRAF6 activation by TGF- β in vitro in A20^{-/-} MEFs. These results implicated TRAF6 and TAK1 as additional targets for A20 anti-fibrotic activity in fibroblasts.

2) The authors propose that DREAM suppresses A20 expression in the context of SSc. How do A20/TNFAIP3 SNPs associated with SSc factor into this model? Is the A20-DREAM regulatory network the primary determinant of fibrosis or are there other mechanisms of A20 downregulation in SSc (i.e., epigenetic or post-transcriptional)?

Responses: We thank the Reviewer for this very germane question. We have incorporated that in the discussion (page 18).

To determine how A20 is down-regulated in SSc and the contribution of DREAM, we performed additional experiments, and provide relevant references. Here we show that DREAM represses A20, and knockdown of DREAM in fibroblasts raises cellular A20 levels (Fig. 5). In contrast, we found no evidence to support epigenetic mechanism underlying reduced A20 in SSc. We discussed our findings under Discussion (page 18).

See also pages 12, 19, 37 and Figure 5; Suppl. Fig. 9A

See also our response to R3

3) The IF staining of DREAM in healthy and SSc skin fibroblasts (Fig. 5D) indicate that DREAM is mainly localized in the cytoplasm. This result seems to contradict the authors' model that DREAM suppresses the transcription of the A20/TNFAIP3 gene in the nucleus.

Responses: In most cells, DREAM is localized in both cytoplasm and nucleus.

In explanted SSc skin fibroblasts, we found increased expression and nuclear localization of DREAM elevated compared to healthy controls. Therefore, the data support, rather than contradict, our model that DREAM represses A20 transcription in the nucleus

See page 12 and Figure 5D (right panel)

4) In Fig. 7F, authors should include a blot for total Smad2 as a control. Why is Smad2 phosphorylation decreased in DREAM KO cells? Is this due to A20 overexpression?

Responses: We previously showed (Bhattacharyya et al, ART, 2016) that Smad2 activation in fibroblasts was attenuated by A20. Here we show that A20 is elevated in DREAM KO fibroblasts, supporting the notion that A20 elevation might be responsible for decreased pSmad2 in these DREAM KO cells.

See Page 13, Figure 7F

5) In Fig. 8C, AdipoRon does not appear to robustly induce A20 expression. How does AdipoRon compare with TNF or LPS treatment for the induction of A20 in fibroblasts?

Responses: We find a significant increase in A20 induced by AdipoRon treatment of fibroblasts. We did not perform a full dose-response for AdipoRon, which at higher doses might be a more potent stimulator of A20.

The stimulation of A20 in response to AdipoRon treatment seems to be more durable than that induced by LPS/TNF treatment. These findings are discussed under Discussion.

See Page 20 and Figure 8C

Minor points:

1) It seems that the expression of A20 mRNA in SSc skin is just more variable compared to healthy controls (Fig. 1A).

In response, patient heterogeneity in three public SSc skin transcriptome datasets we analyzed might account for variable A20 data, which nonetheless showed statistical significance compared to controls.

2) Why is there no statistically significant difference in Col1a1 mRNA between WT and A20 fibrocyte mice treated with bleomycin (Fig. 4B?)

In response, we have increased sample numbers indicated in the Figures (dot plots). In this revised Fig. 4B, we showed significant difference in wildtype mice injected with bleomycin. In panel C, we have increased the number of samples to N=4.

Reviewer #2 (Remarks to the Author):

This is a well-designed and -written submission from one of the leading groups in systemic sclerosis. In this manuscript, the authors proposed that ubiquitin-editing enzyme A20 is a main contributor to the susceptibility of developing fibrosis in systemic sclerosis (SSc). The suggested mechanism of A20 in driving fibrosis depends on the levels of its negative transcriptional regulator, DREAM, in tissues of interest. When DREAM is present, the expression of A20 is reduced with an increase of phenotypical characteristics of SSc. The authors demonstrate that A20 expression is reduced in SSc tissue, while DREAM levels are elevated. A20 haplo-insufficient mice confirmed a similar phenotype between the mouse and human SSc skin biopsies. The A20 levels affect survival, differentiation, and activation of fibroblasts, thereby increasing the sensitivity for development of fibrosis. DREAM's regulation of the A20 response uncovers a novel fundamental role for this transcriptional repressor, known primarily as a modulator of pain and inflammation. The authors propose this feedback network as a possible pharmacological target to augment A20 expression and reduce fibrosis in SSc.

Specific points:

1) The methods used for immunofluorescence quantification are not clear and should be explained in detail in material and methods.

In response, IF methods are detailed under Methods.

Page 37

2) Figure 8D is an important figure because it shows the reversal of skin fibrosis. Why is the depicted H&E staining so different, when compared to the other H&E's presented in the paper? Format and labels are also different, they should be consistent with the other figures.

In response, in figure 8, methyl green was used as counterstaining to evaluate target protein localization instead of H&E. As suggested, we reformatted Figure 8 (**Figure 8E**)

3) The authors suggest the use of pharmacological approaches to augment endogenous A20 as a therapeutic approach in SSc. What will be the effect of this intervention in normal tissues? What will be the effect of such treatment in the intensity and duration of other inflammatory responses? The authors should discuss the limitations and caveats of this approach.

In response, we have inserted discussion of limitations and caveats associated with pharmacological augmenting endogenous A20

Page 20

Reviewer #3 (Remarks to the Author):

Wang et al. present a study about the contribution of A20 and DREAM to fibrosis in systemic sclerosis and thus continue previous studies. However, as presented, the study is not convincing and I have major concerns to improve the manuscript.

Comments:

1. Figure 1B: The quality / resolution of the IF pictures should be improved, also the scale bar should be better visible. Quantitation: The authors measured fluorescence intensity, but how were fibroblasts identified, as they give % of positive fibroblasts? In addition, how were unspecific signals identified?

In response, this figure was replaced with improved quality images with scale bar. Fig. 1B shows quantitation of A20+ fibroblasts (Fig. 1B), whereas Fig. 1C shows A20 fluorescence intensity in measured in healthy and SSc fibroblasts.

pages **5 and 23** (legend) and **Fig. 1B**.

2. Figure 1C: What is the exact size of the bar? The legend says 10 μm , the writing in the pictures say 20 μm and 5 μm . This discrepancy can also be found in Fig. 1E (20 μm vs. 50 μm).

In response, we have revised the scale bar.

3. Figure 2A: Instead of labelling the fold change as low and high, it would be better to give fold change values corresponding to the different color shades.

In response, we have revised the labelling to show fold change (Figure 2A)

4. Figure 2B: Why were only three mice per group analyzed by qPCR for Col1a1 and Col1a2, and not the whole groups?

In response, we have included additional samples in our qPCR result (Figure 2B)

5. Figure 2E: The authors used aSMA for identification of activated fibroblasts. However, aSMA is also highly expressed in vessels. How did the authors distinguish fibroblasts and vessels, especially small vessels? The standard deviations in the quantitations are in part quite high. More samples could solve this problem.

In response, we explain that when quantifying myofibroblasts, ASMA+ cells with distinct vessel morphology were excluded from analysis and only cells with characteristic fibroblast spindle shape is counted. Page 22, Figures 3D,E; Fig. 2E

6. Figure 2F: The right trichrome-stained picture (A20+/- Tsk1/+) looks a bit squeezed.

In response, we have modified the images. Figure 2F and Figs. 4B and C

7. Figure 3: In panel A, it would be better to show low magnification of the whole lung section (at best) and then magnify a certain part.

How many samples have been analyzed for each read-out?

Panels D and E: Same comment as for Fig. 2E: How were fibroblasts identified and discriminated from vessels?

In response, we have replaced the figure with low mag images (Figure 3A). The dot blots showed the number samples used for each readout.

Panels D and E: See response to comment 5.

8. Figure 4: Panel B: In contrast to the statement in the text, there are no significant increases for Col1a1 (for all comparisons). In addition, how do the authors explain that there is no increase in wildtype mice injected with bleomycin (the only exception is IL6)? There should be a significant induction at least of collagen expression also in wildtype mice injected with bleomycin. Panel C: According to the dots in the graph, the authors analyzed with an N of 2 and 3? This is statistically not convincing. For the immunofluorescence staining, again a cell type marker to discriminate aSMA-positive fibroblasts from vessels is missing.

In response, we have increased sample numbers indicated in the Figures (dot plots). In this revised Fig. 4B, we showed significant difference in wildtype mice injected with bleomycin. In panel C, we have increased the number of samples to N=4.

Figs. 4B and C

9. Figure 5, panel C: There is again a discrepancy about the bar size: 50 μm in the legend, 20 μm or 10 μm in the picture.

In response, we have revised the scale bar.

10. Figure 6. Panel B and C: Again $n = 3$ with high standard deviations are statistically not convincing. Panel E, and Fig. 7D: Same as for previous figures and stainings, how were fibroblasts identified and discriminated from vessels?

In response, we increased the number of samples of 6B and 7B ($n=4$).

Figs. 6B and 7B

Fig. 7D: See response to Figure 2E.

11. In general, the sample numbers in many read-outs are very low with $n = 3$ or even $n = 2$.

In response to the Reviewers' general comments, we have performed additional experiments and tried to increase the sample numbers wherever possible as indicated in Figures (1C, 2B, 4B, 5D, 6B and C, and 7B).

12. The mechanism of DREAM upregulation in SSc is missing and should be analyzed.

Response: We thank the Reviewer for this valuable suggestion. To address this question, we performed a series of new experiments. First, we determined the impact of TGF- β on DREAM expression, and found no significant change (data not shown). Next epigenetic mechanisms underlying DREAM upregulation in SSc were evaluated. These new experiments, which suggest a potential role for histone acetylation driving DREAM upregulation in SSc fibroblasts, are shown.

Pages 12, 19, 30, 31, 36 and Suppl. Figs. 9B, C

13. To proof whether DREAM is suppressing A20 by direct binding to the promoter in SSc can be easily evaluated by chromatin immunoprecipitation in SSc fibroblasts.

Response: We thank the Reviewer for this valuable suggestion. Due to the unavailability of ChIP compatible anti-DREAM antibodies we were unable to perform ChIP assays, as suggested. However, we have shown previously that DREAM binds directly to A20 DRE to regulate A20 expression in a variety of cell types (Tiruppathi et al, Nat Immunol. 2014). An alternate approach used to address this question involved DREAM knockdown in SSc fibroblast, which augmented A20 expression and reduced collagen and ASMA levels, indirectly supporting the notion that DREAM is controlling A20 in SSc fibroblasts by transcriptional repression.

Pages 12, 19, 30, 31, 36 and Suppl. Fig. 9A

14. The authors revealed autoantibodies in the A20 +/- mice. Did they investigate the contribution of B cells and other leukocytes to the higher susceptibility of these mice to fibrotic stimuli?

Response: We did not directly address the role of B cells in spontaneous antibody production, but discuss the findings and potential role of B cells in the context of a publication

See **page 17** and **Fig. 2A**

15. In the methods section, the authors should explain how they analyzed the genome-wide expression profiling in more detail. Which software and packages were used? Did they only use one software/package? How did they generate the plots?

In response, we included the description under Methods.

Page 36

Reviewers' Comments:

Reviewer #1:

Remarks to the Author:

The revised manuscript is improved and most of my previous comments were satisfactorily addressed by the authors. However, two specific points still need to be addressed:

1) The new TRAF6-Ub blot in Supplementary Fig. 6B is of poor quality with a high background signal. This experiment should be repeated to obtain better quality images. Is the TRAF6 blot from the IP or input from the lysates? It is important to have an IP control to ensure similar levels of TRAF6 were immunoprecipitated in each sample. IL-1 or LPS treatment could serve as a positive control for TRAF6-Ub.

2) Total SMAD2 blot is still missing in Fig. 7F. This is needed as a control for proper interpretation of the pSMAD2 blot.

Reviewer #3:

Remarks to the Author:

The authors Wang et al. addressed the comments satisfactorily. I have no more concerns to be addressed.

REVIEWER COMMENTS

Reviewer #1 (Remarks to the Author):

The revised manuscript is improved and most of my previous comments were satisfactorily addressed by the authors. However, two specific points still need to be addressed:

1) The new TRAF6-Ub blot in Supplementary Fig. 6B is of poor quality with a high background signal. This experiment should be repeated to obtain better quality images. Is the TRAF6 blot from the IP or input from the lysates? It is important to have an IP control to ensure similar levels of TRAF6 were immunoprecipitated in each sample. IL-1 or LPS treatment could serve as a positive control for TRAF6-Ub.

We agree with the reviewer's comment. We, therefore, repeated the experiment and present the results. Accordingly, we have replaced the previous blot with results from the new experiment with much cleaner images with appropriate controls. We have revised the Materials and Methods and Results section to reflect these changes on Page 10 (result section), 29 (legend) and 36 (Method) highlighted and underlined.

2) Total SMAD2 blot is still missing in Fig. 7F. This is needed as a control for proper interpretation of the pSMAD2 blot

In response, we have included total Smad2 blot in this revised version.

Reviewers' Comments:

Reviewer #1:

Remarks to the Author:

The revised experiments addressed my previous concerns.